# Use of Bio-Epoxies and Their Effect on the Performance of Polymer Composites: A Critical Review

**DOI:** 10.3390/polym15244733

**Published:** 2023-12-18

**Authors:** Monica Capretti, Valentina Giammaria, Carlo Santulli, Simonetta Boria, Giulia Del Bianco

**Affiliations:** 1School of Science and Technology, Mathematics Division, University of Camerino, Via Madonna delle Carceri 9, 62032 Camerino, Italy; monica.capretti@unicam.it (M.C.); valentina.giammaria@unicam.it (V.G.); simonetta.boria@unicam.it (S.B.); giulia.delbianco@unicam.it (G.D.B.); 2School of Science and Technology, Geology Division, University of Camerino, Via Gentile III da Varano 7, 62032 Camerino, Italy

**Keywords:** bio-based thermosets, lignin, cardanol, vegetable oils, natural fibers, mechanical properties, impact properties, thermal properties, glass transition temperature

## Abstract

This study comprehensively examines recent developments in bio-epoxy resins and their applications in composites. Despite the reliability of traditional epoxy systems, the increasing demand for sustainability has driven researchers and industries to explore new bio-based alternatives. Additionally, natural fibers have the potential to serve as environmentally friendly substitutes for synthetic ones, contributing to the production of lightweight and biodegradable composites. Enhancing the mechanical properties of these bio-composites also involves improving the compatibility between the matrix and fibers. The use of bio-epoxy resins facilitates better adhesion of natural composite constituents, addressing sustainability and environmental concerns. The principles and methods proposed for both available commercial and especially non-commercial bio-epoxy solutions are investigated, with a focus on promising renewable sources like wood, food waste, and vegetable oils. Bio-epoxy systems with a minimum bio-content of 20% are analyzed from a thermomechanical perspective. This review also discusses the effect of incorporating synthetic and natural fibers into bio-epoxy resins both on their own and in hybrid form. A comparative analysis is conducted against traditional epoxy-based references, with the aim of emphasizing viable alternatives. The focus is on addressing their benefits and challenges in applications fields such as aviation and the automotive industry.

## 1. Introduction

By the name of ethoxyline resins, epoxy resins were patented by the German P. Schlack in 1934. Since then, they have gained growing significance across a wide range of applications. These polymers consist of two components: hardner and resin which exothermically react. The epoxy synthesis is made in two phases: the monomer formation and the polymerization. The latter phase includes the harderner use, commonly the epichlorohydrin (ECH), which reacts with the bisphenol A (BPA) producing the bisphenol A diglycidyl ether (DGEBA). Despite the most common type of epoxy resin is the DGEBA-based epoxy resin (Figure 1), Bisphenol F or brominated can be used instead of BPA. The resulting epoxy polymers are more chemically resistant and less viscous, since for the same weight they have more epoxy groups than BPA.

The epoxy content, quantified by the epoxide equivalent weight (EEW), representing the quantity of epoxy resin grams required to produce 1 mole of epoxy groups, serves as a pivotal parameter in assessing the mass of hardening or curing agents to optimize the polymerization process. The ultimate properties of thermosetting materials are contingent upon the specific combination of hardening agents and epoxy resins. For instance, the glass transition temperature of cured resins strictly depends on the molecular structure of the hardening agents. The hardening process, indeed, significantly augments the final properties of the resin, encompassing mechanical and thermal attributes. Notably, cure at lower temperatures, such as at room temperature, results in a progressive increase over time in both tensile strength and elastic modulus [1].

Nowadays epoxy resins constitute a preeminent class of thermosetting polymers extensively employed in different fields including engineering [2], electronics [3], civil construction [4], and biomedical industries [5]. This widespread adoption is attributable to their exceptional mechanical attributes, chemical and water resistance, long-term robustness, adaptability, and facile manufacturability. In composite applications epoxy can be used either as filler (or modifiers) to enhance reinforcements of polymer composites or as thermoset matrix reinforced with (synthetic or natural) fibers [6].

Despite the great reliability guaranteed by epoxy thermosets also in high-performing applications (like in aerospace [7] and automotive fields [8]) their use presents some drawbacks. First of all, most of the epoxies come from fossil feedstocks. The majority is highly toxic, such as BPA and ECH, which are carcinogenic and mutagen endangering the human health [9,10]. Although chemical recycling of epoxies has also been investigated [11,12,13], it is proved still quite difficult. Once the hardening process is done, it is impossible to separate the components again, which hold together thanks to strong chemical bonds. Epoxy resins can be disposed of through pyrolysis, a high energy-consuming process, which however under high temperature degrade them into unrecoverable fuel. Alternatively, it is possible to modify the critical parts of such composites to make them recyclable, for instance transforming the hardener component. In this regard, it is worth mentioning Recyclamine^TM^ (by Connora Technologies) as a new class of high-performance amine-based epoxy curing agent. Such hardeners, thanks to enabling cleavage points at crosslinking sites, allow the transformation of thermoset epoxy into a thermoplastic state [14].

On the other hand, focusing on sustainability concerns, it is crucial to find green alternatives to petroleum-based epoxy, where DGEBA is fully or partially replaced by bio-based and environmentally friendly materials [15]. The following sections, indeed, concentrates on outlining potential principles suitable for synthesizing bio-based epoxy resins, and their use in composite applications. In this regard, Section 3 was dedicated to the analysis of the mechanical and thermal properties of composites reinforced with bio-based epoxy resins. Particular attention was paid to purely natural composites since bio resins have been developed following the increasing widespread of vegetable fibers. However, there are several issues linked to the use of bio-resins in composite applications: fiber-matrix adhesion, humidity, and water absorption. All these aspects should be addressed according to the reinforcement type. At the current state of research, it is particularly difficult to use 100% bio-composites in structural applications. The poorer mechanical performance achieved by natural fibers with respect to synthetic ones, especially carbon, still limits their applicability. Hybridization [16,17,18], in this sense, should be considered a viable solution to exploit the potential of natural reinforcements, aiming at reducing the carbon footprint, while preserving, as much as possible, the mechanical properties of the composite. Differently from automotive and transportation sectors, there are other fields in which bio-epoxy composites were already successfully used (see Section 4). Referring to the applications, nanocomposites could also be mentioned. However, since they fall beyond the aim of our research, we limit to cite here a recent review [19] on the analysis of different bio-epoxy nanocomposites and their possible applications in the near future.

The paper is structured as follows (see Figure 2). Section 2 addresses the principles employed in synthesizing new bio-epoxy systems from bio-feedstocks, along with a brief overview of commercially available solutions. In particular, non-commercial resins derived from wood (Section 2.2.1), food waste (Section 2.2.2), and vegetable oil (Section 2.2.3) are comprehensively discussed. Section 3 is dedicated to presenting bio-epoxy applications in composites, distinguishing between the types of reinforcement used, specifically, synthetic or hybrid (Section 3.1) and purely natural (Section 3.2). Future challenges and potential research directions to optimize the performance of bio-epoxies are discussed in Section 5. Finally, Section 6 encompasses the proposed applications of innovative bio-epoxies and those enhanced with bio-fillers, followed by concluding remarks.

## 2. Bio-Epoxy Resins

Based on the aforementioned observations, the recent surge in the demand for sustainability has prompted a heightened effort of researchers and industries towards utilizing bio-derived feedstocks as a substitute for DGEBA. Epoxy resins, indeed, can be synthesized from various types of bio-based materials. The chemistry of epoxy thermoset itself suggests the type of sources to select as feedstocks. Wood takes center stage due to its exceptional resistance to heat and its non-melting characteristics. Comprising primarily cellulose, lignin, and water, each with its distinct melting point, wood, when subjected to heat, undergoes a transformation, yielding diverse substances like charcoal, methanol, and carbon dioxide. Thus, *lignin* can be regarded as a viable environmentally friendly substitute for BPA, serving as a source of phenolic compounds. The content of these compounds is not easily predictable through different extraction processes, as reported in the review by Lu and Gu [20]. Lignin is readily accessible, ranking second only to cellulose as the most abundant organic material on Earth, and can be extracted from various sources such as wood, cotton, jute, hemp, and black liquor. Identifying the source and extraction method is crucial, as it inevitably influences the structure and properties of the final product, as discussed later. Hence, lignin, due to its abundance, sustainability, and inherent polyphenolic structure, has been regarded as a promising renewable raw material for bio-based epoxy resins [21,22,23,24,25,26,27,28]. In addition, lignin depolymerization offers a route to generate alternative bio-based thermosetting systems, as those derived from *vanillin* [29,30,31]. Cellulose or lignin can be utilized also to derive *resorcinol* [32,33,34], a phenolic compound from which ensuing polymers find applications as adhesives, coatings, plastic moldings, and in the formulation of rubber composites. Furthermore, various types of wood, including oak, cedar, walnut, and select mahoganies, naturally contain *tannic acid*, a water-soluble polyphenolic compound with a substantial molecular weight. This compound can be employed as a bio-curing agent or epoxy monomer in epoxy systems [35,36,37,38,39] as well as *rosin*, which is another important abundant natural product extracted from conifer trees [40,41]. In Section 2.2.1, a comprehensive analysis is conducted on epoxy systems derived from the previously mentioned bio-feedstocks.

One of the notable alternatives to BPA among bio-aromatic compounds is *cardanol*, a brown liquid extracted from cashew nut shell liquid (CNSL). Given the substantial scale of cashew nut production, particularly in Africa and Asia, such non-edible compounds constitute a notably abundant resource in nature. A cardanol-derived epoxy (see Section 2.2.2) formulation can be synthesized through either the epoxidation of cardanol or its utilization as a curing agent [42,43,44,45,46,47]. Moreover, bio-content can be integrated into epoxy systems using natural polysaccharides, such as *D-glucose* [48]. Further, as food waste-based feedstocks it should be mentioned the *furan*, which presents a compelling potential as substitute for phenyl building blocks derived from petroleum in epoxy thermosets (see Section 2.2.2) [49,50,51,52,53]. This organic aromatic compound, composed of four carbon atoms and one oxygen, can be derived from bagasse, the residual material from sugar cane processing, as well as from corn cobs or other biomass sources. Additionally, clove, cinnamon, pepper, and turmeric can be considered viable bio-sources for epoxy, especially in the extraction of *eugenol* [32,33,51,54,55] (refer to Section 2.2.2). This phenolic compound is found in the oils of these plants and has demonstrated efficacy as a curing agent, flame retardant, or as an integral component in epoxy monomers. Epoxidation reaction also offers a possible pathway for incorporating natural oils into bio-epoxy resin formulations. Among edible oils, options such as *soybean* [56,57,58,59,60], *linseed* [59,61,62,63] and *hemp* [60,64,65], stand out as potential sources for developing environmentally sustainable epoxy systems, as extensively analyzed in the comprehensive overview by Mustapha et al. [66]. From triglyceride vegetable oils, such as coconut, soybean or palm oil, it is possible to obtain the *glycerol*, which is also commercially used to produce commercial resins [67,68,69] and proposed in the literature for the synthesis of new epoxy systems [38,70,71]. Furthermore, exploration into the integration of non-edible oils such as *castor* [72,73], *karanja* [37], and *canola* [74]. A detailed description of the incorporation of these vegetable oils into epoxy systems can be found in Section 2.2.3.

In Section 2.1 and Section 2.2, an investigation is conducted on commercial bio-epoxy resins and those independently synthesized from their individual constituents, respectively, based on the methods and types of raw materials employed. Considerable emphasis is directed towards the latter, with the objective of discerning novel bio-epoxy resins as viable substitutes for petroleum-derived counterparts. This provides the opportunity to also address the synthesis procedures of these eco-friendly resins, highlighting weaknesses and strengths for potential improvement in future advancements. Lastly and most importantly, the following overview takes into consideration literature works that address epoxies containing a substantial proportion of bio-derived components, i.e., *at least 20%*.

### 2.1. Commercial Resins

Despite the demand for eco-friendly resins being quite recent, driven by growing concerns about environmental sustainability, several companies have already invested resources in green innovation. Their objective is to narrow the performance gap between bio-based epoxy systems and their fossil fuel-derived epoxy counterparts, allowing them to effectively compete with traditional materials. In recent years, some of the main players which gained international relevance within the bio-based epoxy resin market are Gurit, Sicomin, Entropy Resins, Cardolite Corporation, Ecopoxy, EasyComposites, and Huntsman Corporation.

Table 1 shows different commercial bio-based epoxy resins, each possessing a bio-content exceeding 29%. Several of these are subjected to detailed analysis within the composite applications as outlined in Section 3. The tensile and flexural mechanical properties compared with a DGEBA-based counterpart—i.e., the Araldite LY556 and the hardener Aradur 917—are presented in Table 2, beginning with the maximum bio percentage. Henceforth, all the bio-resins reported in tables are systematically compared to a non-bio counterpart.

### 2.2. Non-Commercial Resins

The upcoming sections focus on the synthesized epoxy systems sourced from different bio feedstocks found in the literature. In particular, these thermosets are described based on the type of bio resource from which they originate: wood-based (Section 2.2.1), food-waste based (Section 2.2.2), and vegetable oil-based (Section 2.2.3). For the sake of clarity, all referenced samples, hereinafter summarized in tables and figures regarding their thermomechanical properties, are abbreviated using an acronym of the used components. The numerical value preceding this code indicates the percentage of these materials in the epoxy, as well as for the curing agent when that value is available. Within the text, when the hardener used is bio-based, it is explicitly declared. Otherwise, its origin from non-bio sources is implied.

#### 2.2.1. Wood-Based

##### Lignin

Along with cellulose, *lignin* constitutes the primary component of wood, so it is worth addressing a thorough discussion on how it serves as a valuable green raw material for epoxy systems. The extraction technology identification is crucial to determine the final structure of lignin. Basically, two main categories of processes can be distinguished in industrial or laboratory-scale extraction procedures. The extraction of lignin from the pulp and paper industry encompasses various technologies, including widely utilized methods such as kraft, organosolv, and hydrolysis. This yields different types of lignin, namely kraft lignin (KL), organosolv lignin (OL), and hydrolysis lignin (HL). Pre-treatment methods, categorized as physical, chemical, or biological, can also be employed. The process conditions and variations may result in heterogeneous compounds from the same feedstock, influencing composition and purity [83]. Epoxidation, depolymerization, and glycidylation are the most effective treatments commonly performed on pristine lignin. The typically unknown nature of such raw materials, particularly when classified as waste derivatives, emphasizes the necessity for additional research in this domain to gain a more comprehensive understanding of how chemical constituents impact the properties of bio-epoxy resins.

Hereafter, available studies in the literature regarding the incorporation of various types of lignin into epoxy systems (extensively described in Table 3) will be analyzed, and the results will be compared with those of traditional epoxy thermosets.

Bagheri et al. [21] introduced a completely bio-derived substitute for BPA epoxies, integrating bamboo fibers in composite applications (see Section 3.2). This was achieved by utilizing an epoxidized softwood KL (EKL), with an average molecular weight ($M_{w}$) of 9300 g/mol, in combination with ECH derived from glycerol. Additionally, bio-based curing agent (NT1515 supplied by Cardolite Co., Bristol, PA, USA) sourced from cashew nut shell was employed, resulting in a bio-epoxy system with higher viscosity (21 ± 0.5 Pa·s) compared to the commercially used EPON 828 epoxy (15 ± 0.4 Pa·s). That increase can be attributed to the inherent high viscosity, molecular weight, and branched structure of the lignin resin, consequently resulting in a reduction in crosslinking density. After curing, lignin-based resin presented a lower glass transition temperature ($T_{g}$) than the corresponding EPON 828 as determined from the peak of tanδ in dynamic mechanical analysis (DMA). This outcome implies a potential enhancement in dissipated energy under stress when lignin is incorporated. On the other hand, from the bending test, a slight reduction in flexural performance was observed in neat bio-resin compared to the commercial BPA-based one. However, the slight deviation of bio-resin from the recommended viscosity range did not lead to a loss of the final mechanical properties. This is because the potential compromise in wettability and processability of composites was offset by the good compatibility between the fibers and the matrix. Furthermore, lignin-based epoxy demonstrated lower thermal stability (200–500 °C) compared to EPON 828 (300–450 °C). In natural fiber-reinforced composites, this reduction is not a significant concern, given fibers usually decompose at lower temperatures than the resin. The authors highlighted, through thermal gravimetric analysis (TGA), a broad decomposition range in the case of lignin-based epoxy. This is attributed to a higher variation in $M_{w}$ of the resin, resulting in heterogeneous behavior, in contrast to the commercial DGEBA-based epoxy.

Engelmann et al. [22] incorporated different contents of pine KL ($M_{w}$ = 1300 g/mol) commercialized by Meadwestvaco (Indulin AT), ranging from 20 to 50 wt%, along with a 1,3-glycerol diglycidyl ether (DGE) and a bio-based curing agent, pyrogallol (PY), the chemical structure of which is depicted in Figure 3. Subsequently, green epoxy composites reinforced with cellulose fibers were obtained from this mixture.

According to the curing agent contents, different effects on the thermomechanical behavior of the bio-epoxy system can be observed. Tensile strength reached the maximum value (82 MPa) along with a modulus of 3.6 GPa, for a 3.67:1.6:1 weight ratio of DGE, lignin and PY, respectively. The presence of the PY agent impeded the reduction in the crosslink of DGE caused by lignin incorporation (25.3 wt%). With the increase in lignin content and PY incorporation, the crosslinking densities raised so that there was a corresponding increase in the $T_{g}$ value (as depicted in Figure 4), even in the absence of PY. This contrasts with the findings in [21] for lignin with a higher $M_{w}$. PY resulted in being crucial to enhance the crosslinking densities of DGE and damping the effect of lignin.

Also, the average storage modulus (E’) distinctly experiences the positive effect of PY, showing and increasing until 4.6 GPa. This value is markedly higher than the one reported in the aforementioned study.

In [23], a hardwood KL was epoxidized (from pulp and paper) with ECH (ECH/KL ratio was 1:3) and then blended with a petroleum-based epoxy (LR200 + LE20). EKL did not show glass transition, resulting in a low thermal stability. Anyway, it is advisable to incorporate EKL rather than KL into epoxy systems, as even at relatively high concentrations, it should not adversely affect the ultimate mechanical performance of the thermoset. The impact strength tested through the Izod impact resulted in being comparable or increased by the addition of 15 or 30 wt% of EKL (2.7 and 2.2 kJ/m${}^{2}$ respectively) with respect to a DGEBA-based resin (2.2 kJ/m${}^{2}$) thanks to the enhanced lignin reactivity obtained through the addition of an epoxy group in the epoxidation. Such a reaction resulted in a favorable adhesion effect as confirmed by shear tests.

Liu et al. [24] synthesized bio-polymers by blending depolymerized kraft lignin (DKL) obtained from Poplar paper black liquor (where $M_{w}$ = 2437 g/mol) and ECH with epoxied cardanol glycidyl ether (ECGE) and methyltetrahydrophthalic anhydride (MeTHPA) as a curing agent. This was undertaken with the aim of mitigating the elevation in viscosity and brittleness induced by lignin, thereby enhancing processability in potential manufacturing processes. The DKL content in the blend ranged from 20 to 80 wt%, and it was compared with a fully DKL-based thermoset. It is noteworthy to highlight that in the blend, the highest lignin content correlates with the highest levels of tensile and flexural strength, as well as the $T_{g}$ value, all of which are comparable to those observed in DGEBA-based epoxy and other previously examined KL-based thermosets. In contrast, pure DKL exhibited a notable decrease in both tensile and flexural strength, approximately 30 and 34%, respectively, when compared to such a blend due to the absence of ECGE. Figure 4 and Figure 5 depict comparable KL-based epoxy systems discussed thus far, highlighting their thermomechanical properties.

The significant difference in flexural strength exhibited by the same amount (100%) of KL in [21,24] (14 vs. 86 MPa) could be mainly attributed to the different proposed curing agent and treatment of lignin. The partial depolymerization of KL [24] might result in a higher epoxy content compared to the epoxidation method on pristine KL suggested in [21], leading to a higher crosslinking density (hence epoxy content) and consequently to increased flexural strength.

As previously noted, it is essential to also conduct comparative analysis of the properties exhibited by the ultimate lignin-based thermoset obtained through different extraction methods. Ferdosian et al. [25] performed essential comparative analysis on lignin-based thermosets from different extraction methods. Blending epoxidized DKL (EDKL) and DOL (EDOL) with a BPA-based epoxy (Araldite^TM^ GZ 540 × 90), and a 4,40-diaminodiphenylmethane (DDM) as curing agent revealed that molecular mass differences affect reactivity. This led to a decreased epoxy content (6 vs. 8 wt%) and lower mechanical performance for EDKL-based epoxy resins. Additionally, an increase in lignin content up to 50 wt% led to a decrease in $T_{g}$, differing from higher $M_{w}$ lignin combined with PY in [22]. On the other hand, as the lignin content exceeded 50 wt%, a notable increase was observed specifically for DKL. This can be attributed to etherification and a higher concentration of hydroxyl groups, enhancing the curing process [27,84].

Beech wood (*F. sylvatica* L.) chips were used as source of OL by Over et al. [26], which were glycidylated (GOL) and then cured with DGEBA and isophorone diamine (IPDA) as a crosslinker. The lignin content was tested at up to 33 wt%, for which the highest $T_{g}$ (138 °C) was recorded accompanied by a tensile modulus of 1.9 GPa. As for the previously mentioned epoxidation of KL, here, the glycidylation of OL led to increase the epoxy content (3.2 mmol/g) of pristine lignin, which was useful to reduce the stiffening effect of increasing the lignin incorporation and especially to enhance the crosslinking densities. Moreover, GOL possessed a greater thermal stability than OL, as reported by TGA, since an increase in $T_{d5\%}$ from 233 to 269 °C was recorded. This assertion finds corroboration in additional experimental investigations pertaining to the thermal degradation of modified KL and OL [85,86]. Given that glycidization and depolymerization have manifested a favorable impact on the thermal degradation of unaltered lignin, it is recommended to undertake such modification processes to enhance the ease of manufacturing and processing the bio-polymer.

The use of epoxidized hydrolysis lignin (EHL) in epoxy thermosets and the obtained thermal properties was investigated by Ferdosian et al. [28]. The work focused on the development of a bio-epoxy system using epoxidized de-polymerized hydrolysis hardwood commercial lignin (EDHL), sourced from Aspen wood and possessing a $M_{w}$ of 2100 g/mol, cured with DDM. Manipulating the lignin content from 25 to 100 wt% led to an increase in the thermoset viscosity of epoxy. On the other hand, raising the DHL-epoxy percentage up to 100 wt% and consequently reducing the epoxy content adversely impacted the thermal performance of the resulting epoxy system. Notably, a reduction in the maximum decomposition temperature ($T_{max}$) and temperature at 5 wt% loss ($T_{d5\%}$) was observed for the increasing DHL-based epoxy content, as well as with respect to the DGEBA-based reference (specifically, the commercial Araldite^TM^ GZ 540 × 90, DGEBA). These results are not unexpected since HL was here characterized by hemicellulose and cellulose, which thermally weakened the epoxy system due to their decomposition at a lower temperature than the DGBEA-based one [87] (see Table 4).

EDKL and EDOL then EDHL were used as matrix plain weave glass fibers (GFs) in reinforced composites by Ferdosian et al. in [25,28], respectively. On the basis of the previous remarks, from the latter study, it was expected that the substitution of DGEBA at certain percentages would lead to a deterioration in the tensile and flexural strength (especially for 75 and 100 wt% of DHL) due to poor glass–matrix adhesion. On the other hand, the 50 wt% of DHL guaranteed the highest tensile strength, which was slightly lower in relation to composites based on DGEBA-DDM. The corresponding modulus values were in accordance with those of DGEBA-based composites as examined subsequently in the context of composite applications.

Across all discussed studies which investigated the potential of bio-epoxy natural fiber-reinforced composites, scanning electron microscope (SEM) analysis consistently demonstrated excellent adhesion between the fiber and matrix, as further examined in Section 3. However, in the context of synthetic fibers, this phenomenon might not be replicated to an equivalent extent. This additionally validates the mechanical performance improvement fostered by lignin incorporation on epoxy systems, when compared to its petroleum-based counterpart in composite.

##### Other Bio-Sources

Starting from lignin, it is possible to derive other bio-sources for green epoxies such as *vanillin*. This aromatic phenolic compound is usually extracted from pulp lignin through oxidation [88,89,90]. The ultimate chemical structure strictly depends on the source of the extraction as already observed for the lignin itself. Nonetheless, this compound proves to be effective in bio-epoxy production due to its functional groups, which include aldehyde, ether, and hydroxyl. These groups play a pivotal role in facilitating reactivity with epoxy groups and expediting the curing reaction.

As remarked in the introductory part of the work, the crosslinking process to which an epoxy is subjected during curing makes it not recyclable through the separation of the resin constituents. The impossibility of reprocessing such type of systems has led to the development of a new class of epoxies, i.e., epoxy vitrimers—inherently recyclable, self-healing, malleable, and dimensionally stable—thanks to the introduction of exchangeable dynamic covalent bonds [91]. The widespread adoption of these innovative systems in applications like adhesives, coatings, and advanced composites, as proposed in the thorough review by Zhang et al. [92], faces challenges due to issues such as the substantial energy demand for their reprocessing and repair. Reprocessability might be further enhanced by providing the desired activation energy through the addition of specific dynamic covalent bonds. In recent years, bio-based epoxy vitrimers have been developed from various bio-feedstocks [93,94,95,96,97,98].

Yu et al. [29] produced a fully bio vanillin-based epoxy vitrimer with possible processability after curing and good acid degradability. IPDA was used as a hardener in the curing reaction with the mono-glycidyl structure of vanillin (Van-Ep) and commercial E51 resin as a reference petroleum-based epoxy. From DMA and DSC analysis, a little reduction in $T_{g}$ was recorded for bio-epoxy vitrimers (see Table 5), confirmed also by a lower crosslinking density than E51/IPDA, due to the monoglycid structure of vanillin, which in turn determined a higher gel and swelling ratio of bio-vitrimers. Van-Ep/IPDA showed a reduced flexibility compared to E51 epoxy, also confirmed by its higher E’. This observation aligns with the effects noted earlier in relation to lignin incorporation. Regarding the mechanical performance of bio-polymers, the tensile test revealed a Young’s modulus comparable to that of the DGEBA-based counterpart, even after undergoing multiple cycles of hot reprocessing post-curing (Van-Ep-IPDA-1, Van-Ep-IPDA-2, Van-Ep-IPDA-3) as shown in Table 5. This aspect holds significant importance in assessing the practical feasibility and application of vanillin epoxy vitrimers.

In a subsequent study [30], the same authors investigated the effect of three other curing agents on vanillin-based epoxy. In addition to IPDA, the use of DDM, IPDA-polyetheramine D230 (IPDA-D230), and diethylenetriamine (DETA) as hardeners were examined from both mechanical and thermal perspectives. First of all, the vanillin vitrimers (VE) presented different $T_{g}$ according to the specific incorporated amine agent. The rigidity and high crosslinking density of DDM, indeed, provided the highest $T_{g}$ (143 °C) that clearly resulted in an increased tensile modulus and lowered elongation at break. Compared with a DGEBA-based epoxy, the IPDA agent ensured a slight greater Young’s modulus but smaller tensile strength, even if it was the highest among all tested vitrimers as shown in Table 6.

The addition of D230 to IPDA enhanced the flexibility of the final thermoset due to the inherently lower crosslinking density of the epoxy. However, this modification significantly impacted the tensile strength of the neat thermoset. Nevertheless, when carbon fibers are integrated with such matrices, the IPDA amine proved to be the most effective curing agent. Despite the mechanical superiority of VE/DDM, the corresponding composite samples in this case exhibited additional defects attributed to poor adhesion with the reinforcement worsening by a powder surface. Hence, in view of a composite application, it is crucial to evaluate not only the mechanical performance of neat resin but, more importantly, the compatibility of such matrices with the selected reinforcement. In this case, while VE/IPDA-D230 provided the best interfacial bonding, the lower mechanical performance compared to VE/IPDA led to the latter being preferred as the matrix for reinforcing carbon fibers. For further improvements, it may be of interest to explore chemical grafting [99], crystallization [100] techniques, or the incorporation of nano-reinforcements [101]. These approaches aim to improve the wettability between the fiber and matrix, thereby maximizing the mechanical performance of the neat resin. Nabipour et al. [31] conducted the synthesis of a fully bio-epoxy resin with a reduced flammability characteristic, using a vanillin-derived Schiff base epoxy monomer (VTA-EP), which was cured with a bio-based 5,5’-methylenedifurfurylamine (DFA). The resulting bio-epoxy system exhibited superior thermal stability and mechanical performance compared to the reference DGEBA-based epoxy (EEW = 210–244 g/eq) cured with DDM. Once again, an increase in crosslinking density determined the rising of $T_{g}$ with respect to DGEBA-DFA. Thanks to the Schiff base, the flame retardancy of the epoxy system improved as resulted from the smaller $T_{d10\%}$ and $T_{max}$ compared to DGEBA/DFA as recorded by TGA. At 500 °C, the VTA-EP/DFA composite demonstrated a higher char yield compared to DGEBA/DFA, with values of 59.4 and 33.3%, respectively. This observation indicates a greater capacity for carbonization. Moreover, cone calorimetry revealed a decreased flammability in bio-based polymers compared to traditional ones. Finally, the vanillin incorporation induced an improvement also of some mechanical properties of the epoxy system since either elongation at break (due to greater rigidity) and tensile strength increased with respect to DGEBA/DFA.

The latest discussed studies proposed new fully bio epoxy solutions, which involved the replacement of DGEBA with renewable natural resources and encompassed also bio-based curing agents. In this context, also Mattar et al. [32] conducted experimental tests on autonomously synthesized fully bio epoxies derived from *resorcinol*, a phenolic compound (see Figure 6) that is soluble in alcohol, ether, and water. This involved combining resorcinol diglycidyl ether (RE) with diamine-allyl eugenol (DA-AE) or diamine-limonene (DA-LIM) as bio-hardeners, or curing it with hexamethylenediamine (HDMA). This process yielded three distinct bio-based epoxy systems: fully bio RE/DA-AE and RE/DA-LIM, and partially bio RE/HDMA. Then, the study [33] evaluated the performance of recycled carbon fiber reinforced with such polymers as discussed more in detail in Section 3.1.

With respect to HDMA and DA-LIM, the aromatic group of DA-AE reduced reactivity to epoxy, lowering crosslinking densities and curing enthalpy. This reflected in the $T_{g}$ and $T_{max}$ values from DMA and TGA, with RE/HDMA showing the highest (80 and 353 °C). RE/HDMA, with the lowest molar mass, exhibited the highest values. Substituting RE with DGEBA slightly decreased the thermal parameters, continuing with DA-LIM or DA-AE replacing HDMA. All bio-resins showed flexural performance comparable to the DGEBA counterparts, with strength values ranging from 85 to 99 MPa and moduli from 2.2 to 3.0 GPa (see Table 7). Flexural modulus values were sensitive to the chemical structure of the curing agent, decreasing as the number of atoms between two amines increased. The lowest value was observed for the DA-AE-cured thermoset, with an aromatic structure having the greatest number of atoms (17) between two amines. It is crucial to emphasize that the RE/DA-LIM formulation, which is entirely based on resorcinol and utilizes a bio-curing agent, exhibited the highest flexural performance, surpassing even that of the reference DGEBA/HDMA. Nevertheless, it is imperative to conduct tests on tensile and impact properties to comprehensively characterize and compare the performance of these bio-epoxies with petroleum-based counterparts, with the aim of suggesting viable bio-replacements.

Partially and fully bio epoxy thermosets, replacing DGEBA with RE and using HDMA as the hardener, demonstrated enhanced crack propagation resistance. DA-LIM offered the best compromise between mechanical performance, toughness, and crack propagation resistance. Although the bio-based curing agents negatively affected thermal stability, the addition of recycled carbon fibers mitigated this effect. Notably, the bio-based samples outperformed the DGEBA-based counterparts in fire resistance, crucial for safety in aerospace, automotive, and construction applications.

Desai et al. [34] employed a resorcinol-based epoxy thermoset (with ECH as a curing agent) reinforced with a small proportion of glass, nylon, and polyester fibers (1, 3, 5 wt%) for the production of fiber-reinforced composites through additive manufacturing. Under tensile test, the neat resin exhibited a strength value of 6 MPa and 0.3% as the elongation at break, surpassing that of any tested composite configuration. The $T_{max}$ value recorded by TGA is significantly higher than that reported in [32], which may be attributed to the use of different curing agents and inherent thermal degradation behavior ($T_{ECH,max}>T_{HDMA,max}$).

*Tannic acid* (TA), which can be obtained from a range of woods and leaves, falls under the category of tannoids (or tannins). It is characterized by a large amount of phenol groups, which easily react with epoxy compounds to give phenolic hydroxy ethers. This suggests the possibility to create bio-epoxy based on such tannins. In this context, Kim et al. [35] presented the use of TA and DGEBA (with 0.8, 1.0, 1.2 tested molar ratios) in the synthesis of an epoxy matrix for composite applications with carbon fiber reinforcement, focusing on the resulting flame-retardant advantages as well as in [36]. In these contexts, TA was employed solely as a curing agent, as done by the majority of available literature results [37,38]. Instead, here, the analysis primarily focused on the potential utilization of a bio-hardener to augment the fiber/matrix interface or diminish flammability in fiber-reinforced polymer applications. In [35], the reported interlaminar shear strength (ILSS) values demonstrated that an increase in TA content led to enhanced fiber/matrix adhesion. The most significant improvement (achieved with a 1.2 molar ratio configuration) demonstrated an approximately 25% increase compared to DGEBA (YD-128)/D230. However, it is important to note that an excessively strong interface may promote brittle fracture, potentially impeding the energy absorption mechanism. The highest flexural strength, indeed, was observed at the optimal molar ratio of 1.0, rather than at the maximum ratio.

As observed in the case of resorcinol and vanillin-based epoxy thermosets, here, the incorporation of bio-derived components proved highly advantageous in retarding flammability. Polyphenols, in fact, represent a category of naturally occurring substances with inherent notable char-forming characteristics. Tannins, in particular, are suitable for applications in flame retardancy due to their outstanding char-forming capability [102]. In the study conducted by Fei et al. [36], TA served as a base for an epoxy modifier—i.e., the carboxylic acid-modified tannic acid (TA–COOH)—resulting in elevated values of $T_{g}$, tensile, and impact strength of the obtained epoxy thermoset with respect to the neat resin. Given the previous assessments regarding the stiffening effect of increasing the crosslinking density, since here a reduction was observed as the TA-based modifier content increased, a contradiction emerges. It is worth noting, however, that the inherent aromatic structure of such a compound inhibited such a negative effect on $T_{g}$, resulting in a linear trend of this temperature with the TA content.

Taking into consideration the environmental impact issue and emphasizing the use of bio-epoxy monomers, more relevant is the analysis proposed by Borah et al. [39]. Here, indeed, a new epoxidized TA based-epoxy was synthesized. The tannin content was between 50 and 100 wt% (samples labeled as 50TAE, 75TAE, 90TAE, and 100TAE), and additionally, a poly(amido amine) was used as a bio-based curing agent (BCA). As for the lignin case, here, the epoxidation took the role of enhancing the reactivity so that a greater bio incorporation is possible. In previous mentioned studies, where TA was used as a crosslinker agent, the higher the bio-content, the higher the $T_{g}$ value. However, in this study, where TA assumed a different role, the effects were not in alignment with the previous findings. This discrepancy may be attributed to variations in the content of DGEBA (50, 25, and 10 wt% for 50TAE, 75TAE, and 90TAE, respectively). 50TAE, in fact, exhibited the highest $T_{g}$ value, and no distinct trend in $T_{g}$ evolution was discernible with varying the TA content. In terms of tensile performance, the sample with the highest TA content (100 wt%) demonstrated the highest strength (20 MPa) and the lowest elongation at break (17%), attributed to the stiffening effect resulting from the bio incorporation. However, it is important to note that such a strength value is significantly lower compared to that of a conventional DGEBA-based epoxy thermoset [103] as reported in Table 8.

On the other hand, the impact energy demonstrated a corresponding trend to the tensile strength, displaying enhancement with increased TA content. This phenomenon can be attributed to the heightened density of interactions between the terminal hydroxyl groups. While the examined bio-epoxies displayed modest tensile strength values, which may not advocate for a complete or partial replacement of DGEBA for structural applications, they did present encouraging outcomes for coating and adhesive-based applications. Specifically, the adhesive strength increased with higher TA content for wood substrates (in contrast to aluminum), surpassing that of DGEBA-based epoxy.

*Rosin* (R), the principal component of pine resin, represents another natural feedstock that can be employed in the production of epoxy thermosets. Deng et al. [40] introduced novel epoxy systems, employing rosin as a curing agent. Their objective was to address the significant issue of brittleness that has been discussed in previous studies in the literature when incorporating this compound into epoxy systems. To do this, ethylene glycol diglycidyl ether (EGDE) was employed as flexible chain in epoxy resin preparation.

In [41], R is fully integrated into the epoxy system, functioning both as a curing agent and epoxy monomer. The recorded flexural strength and modulus values, as well as the strength from Izod impact, were slightly smaller than those of the conventional DGEBA-based epoxy. In this regard, it is worth noting that a compelling comparison could be made with another fully bio-based epoxy system examined in this review, specifically the one based on resorcinol, and limonene or eugenol as hardeners. As demonstrated in Figure 7, RE-based systems exhibited superior flexural strength and modulus. Among the options investigated up to this point, such bio-based epoxy types stand out as quite promising, namely in terms of their flexural properties, fully renewable alternatives to fossil fuel-based epoxy systems.

#### 2.2.2. Food Waste-Based

As already mentioned in Section 2, *cardanol* (see Figure 8) is obtained by decarboxylation of anacardic acid, of which CNSL is composed. CNSL is obtained directly from the shell of the cashew nut and can therefore be classified as a food waste-derived product.

As it is at the same time cheap, easily accessible and a very promising product, companies like Cardolite have developed various commercial products, whose properties are collected in Table 1. An attempt to increase the total bio-content of an epoxy resin was made by Iadarola et al. [43], who proposed a mix between two commercial products provided by Cardolite: an epoxy novolac resin NC-547 (84% bio) and a cardanol-based system FormuLITE2502A + FormuLITE2401B (27% bio). The aim is to combine these two compounds to create blends with higher bio-contents. In particular, using the mixture rule, 0, 10, 30 and 50% of NC-547 is added over 100 g of the 2502A + 2401B system to obtain blends with the following bio-content percentages: 27, 31, 41, and 51%. Quasi-static and dynamic mechanical tests are performed to evaluate the variation in the performance results by increasing the bio-content. Before doing this, a DSC analysis was conducted, and the results show a reduction in $T_{g}$ from 86 to 52 °C when passing from a bio content of 27 to 51%. According to the results found by the authors, the increase in bio content determines a reduction in the elastic modulus and strength, while a significant increase in strain is observed for the blend with a bio content of 51%. At the same time, the increasing strain rate causes a reduction in the maximum elongation and an enhancement of the tensile strength and modulus. Finally, SEM micrographs show very good compatibility between the two resins since no defects or phase separations are evident. Moreover, a more ductile fracture is observed, adding bio-content, while a slightly more fragile fracture is noted, increasing the strain rate. Another possibility to exploit the potential of Cardolite products is to use them as flexibilizers to toughen an epoxy novolac resin. This research was conducted by Gour et al. [44,45], who prepared epoxy novolac composites by combining 30% of NC-514 or NC-547 with an epoxy novolac resin. Cardolite NC-514 has a branched chemical structure, while NC-547 has a linear structure, and this difference can influence the performance of the final composites. The first difference is observed in the $T_{g}$ values, which are reduced of 33% and 20%, respectively, if compared to the neat epoxy novolac resin. On the contrary, in terms of the tensile and Izod impact tests, better results are obtained when NC-514 is used, with an increase up to 28% in the mechanical properties. Finally, to enhance the properties of cardanol-derived materials, Darroman et al. [46,47] proposed three different blends obtained by adding epoxy derivatives like sorbitol, isosorbide, and resorcinol to epoxidized cardanol. The aim of their study is to obtain a blend with good thermal, chemical and mechanical properties that allows for coating applications. The authors conclude that it is not easy to replace DGEBA with a single material, but a mixture of bio-based reactants could represent possible success in coatings and also other applications.

Among possible bio-epoxies synthesized from food wastes, sugar-based resins constitute possible alternatives to fossil fuel-based ones. This is due to the presence of hydroxyl groups in the majority of carbohydrates, which facilitate the crosslinking process with epoxy groups in these resins.

Rapi et al. [48] synthesized diepoxide (GP2E, glucopyranoside- based), triepoxide (GF3E and GP3E, glucofuranoside-based and glucopyranoside-based respectively), and tetraepoxy (GP4E, glucopyranoside-based) components derived from *D-glucose*. These compounds were subsequently cured with a DDM hardener, and their thermal stability was investigated. From TGA measurements, the GF4E system showed the same thermal stability of a traditional epoxy resin (ER 1010), due to the resistance of protecting groups in its molecules. The GF3E structure, characterized by its high compactness, resulted in the attainment of the highest recorded $T_{g}$ value at 177 °C, slightly surpassing that of the DGEBA-based epoxy. Furthermore, the author evaluated the potential flame-retardant properties of the synthesized epoxy components, attributing their effectiveness to the substantial char yields observed at 800 °C. These yields exceeded those of ER 1010 in all cases, with the exception of the GP3E system.

Marotta et al. [49] handled a bio-epoxy system based on 2,5-bis[(oxiran-2-ylmethoxy) methyl]furan (BOMF), cured with methyl nadic anhydride (MNA). It was made up of *furan*, which can be synthesized from sugars (e.g., glucose and xylose) via catalytic conversion. The use of maleic anhydride (MA) as a bio-based curing agent, derived from furan and furfural, led to a completely bio-based epoxy resin [50]. The $T_{g}$ value of such bio systems, measured by DSC analysis, resulted in being 29% lower than DGEBA-MA epoxy due to their inherent reduced chain rigidity. The bio-based epoxy compared to the DGEBA-based one still affirmed the superiority as a flame-retardant system due to its higher char yield at 700 °C. Moreover, poor tensile test results indicated that bio-based systems may not be a viable substitute for DGEBA counterparts in composite applications. A significant portion of the studies conducted thus far, indeed, focused on the flexural properties of bio-based epoxy systems, as they exhibited promise and offered potential for practical applications in coatings and adhesives. Here, the lap shear tests, in fact, provided clear evidence of the superior adhesive properties of bio-resins as indicated by stress values that were three times higher than those of the DGEBA-MA resin (see Table 9).

This demonstrated their outstanding adhesion to carbon fiber-reinforced thermosetting plastic (CFRP) substrates through a cohesive failure mechanism, notably different from the adhesive failure observed in the case of the DGEBA-based reference. The authors attributed such adhesive benefits to the presence of hydroxyl groups from the curing reaction of BOMF-MA resins and the stabilizing effect of the furan ring in the interaction with CFRP substrates. Hence, opportunely choosing the curing reagents is imperative for enhancing the adhesive properties of bio-resins, as it directly influences the presence of hydroxyl groups, which, in turn, govern the crosslinking process.

In [51], a very large bio content (93.3%) was used to synthesize a sustainable epoxy resin. In particular, bis(2-methoxy-4-(oxiran-2-ylmethyl)phenyl)furan-2,5-dicarboxylate (EUFU-EP) was obtained from furandicarboxylic acid and eugenol, and cured with methyl hexahydrophthalic anhydride (MHHPA). Despite the lower crosslinking density of EUFU-EP/MHHPA, the combined rigidity of the furan structure led to a greater value of $T_{g}$ compared with DGEBA/MHHPA. Furthermore, flexural tests showed a greater rigidity of EUFU-EP/MHHPA resin than a reference DGEBA/MHHPA from the modulus value (see Table 10), as well as a higher storage modulus at the glassy state. On the other hand, the strength value was slightly lower but comparable with respect to DGEBA/MHHPA. The author stressed the fact that such performance results are in accordance with the high percentage of bio content in the available epoxy resins in the literature. In addition, the dense aromatic structure of furan endowed EUFU-EP with superior flame-retardant properties as evidenced by an about 40% higher char yield at 800 °C than that exhibited by DGEBA-MHHPA.

Wang et al. [52] investigated the flexural and impact properties of an epoxy blend made with commercial furfural-furfuryl alcohol resin (FF) and DGEBA-based epoxy (E51), utilizing various FF-E51 mass ratios (10:0, 6:4, 5:5, 4:6, and 0:10) cured with MeTHPA. The $T_{g}$ value, recorded by DMA, incremented as the epoxy content increased from 40 to 50 wt%, while a decreasing trend was observed when the percentage exceeded 50%, in contrast with the lignin trend influence examined in [22]. Hu et al. [53] observed smaller $T_{g}$ values for a blend of furan (BOF) or phenyl (BOB) diepoxides and DGEBA, as the bio-content increased (0, 30, 50, 70, and 100%) due to the presence of methylene-reduced linkages with furanyl or phenyl segments in BOF and BOB. Nonetheless, comparing the discussed studies related to the use of furan, in [52], employing an equal quantity of FF and E51 yielded the most favorable configuration as indicated by both Charpy and flexural test results. However, it should be noted that these results did not reach the levels reported by Miao et al. [51], who also utilized a higher bio content (as detailed in Table 10).

#### 2.2.3. Vegetable Oil-Based

Some studies examined in Section 2.2.1 have addressed bio-epoxy systems, employing *eugenol* as a curing agent [32,33]. Now, its potential as an epoxy monomer in its own right is analyzed. This natural compound is the predominant component of clove oil.

Chen et al. [54] achieved fully bio-based epoxy solutions by esterifying eugenol with succinyl, adipoyl, suberoyl, and 2,5-furan chloride, coupled with an oxidation of the allylic bond and a self-curing process. This particular curing strategy demonstrated advantages in relation to the rigidity of thermosets and their thermal stability, leading to an augmentation of $T_{5\%}$ compared to a curing reaction involving DDM. Also, Wan et al. [55] synthesized a new flame-retardant epoxy resin from eugenol (DEU-EP) and compared it with a DGEBA-based reference, curing both with DDM. Despite the high bio content (70.2%), the bio-based thermosets presented greater values of storage and loss modulus than DGEBA/DDM (up to the temperature of 97 °C) as well as char yield at 700 °C. This contributes significantly to their efficacy in terms of flame retardancy and carbonization. Notably, here, the self-curing method emerged as the most effective strategy in achieving the highest $T_{g}$ value, even with 100% bio-derived feedstock.

The direct use of vegetable oils is another possible alternative of notable interest to DGEBA due to their versatility, ready accessibility, and renewable nature. *Soybean* and *linseed* oils, for instance, are among the most promising oils applied as monomers in the synthesis of epoxy systems, due to their large content of unsaturated triglycerides. Unfortunately, their poor reactivity still makes DGEBA only theoretically replaceable by such bio feestocks [66]. Furthermore, given the lack of aromatic, aliphatic, or cyclophatic structures that typically characterize the majority of analyzed bio sources thus far, a decrease in the mechanical performance of epoxy systems based on vegetable oils is likely to be expected.

In the work by Pansumdaeng et al. [56], a fully bio-based epoxy system was introduced using soybean oil in conjunction with bio-based crosslinkers, including succinic anhydride (SUC), suberic acid (SU), and sebacic acid (SE). This formulation exhibited high performance in the application of a triboelectric nanogenerator. The crucial role of the epoxidation reaction, as previously demonstrated with lignin, was highlighted in enhancing the reactivity of triglycerides. Consequently, all referenced vegetable oils employed in synthesizing epoxy systems will be considered in their epoxidized form moving forward. The chosen curing agents contributed to an enhancement in the flexibility of the resulting epoxidized soybean oil (ESO)-based thermoset. As a result, although its tensile properties (with strength and modulus values of only 0.2 and 1.5 MPa, respectively) are inadequate for structural composite applications, such bio-epoxy remains well-suited for coating flexible surfaces. Additionally, when resins are utilized as coatings, it is imperative to ensure the hydrophobicity of the epoxy films. In fact, after one day of immersion in water, the ESO films exhibited only a 1% water uptake.

Altuna et al. [57] investigated the effect of increasing the ESO content up to 100% for synthesizing epoxy resin cured with MeTHPA. As the ESO percentage increased, the $T_{g}$ value suffered a decrease due to flexibility enhancing, given by the plasticizing effect of soybean oil incorporation. Furthermore, assessments were conducted on the impact and compression strength to gauge the mechanical capabilities of these bio-thermosets. In terms of impact performance, Charpy tests revealed as the optimum an ESO content of 40%, which yielded a strength value of approximately 0.4 kJ/m${}^{2}$. However, in the case of compression tests, the yield stress and modulus were found to diminish with the increasing ESO content, a trend closely linked to the epoxy content of each respective system, which inevitably decreased with flexible ESO incorporation. The mentioned studies regard the use of a commercial ESO on innovative bio-based epoxy systems. Notably, Zhu et al.’s work [58] stands out, as they blended a commercial epoxy resin (Shell Epon 9500/Epicure 9550) with newly synthesized epoxidized soybean oil derivatives, namely epoxidized methyl soyate (EMS) and epoxidized allyl soyate (EAS). Subsequently, the thermomechanical properties of the resulting thermosets, obtained through a two-step curing process with para-amine cyclohexylmethane (PACM), were compared to those achieved using a commercial ESO. As detailed in Table 11, the commercially available ESO demonstrated mainly inferior performance in almost all cases with respect to the newly synthesized EAS. This was evident in terms of the tensile and flexural strength, as well as the flexural modulus, which were additionally comparable with those of the reference commercial epoxy. Moreover, even if the $T_{g}$ value trend with bio content reported by Altuna et al. [57] is corroborated here, the heightened rigidity of EAS in comparison to EMS and commercial ESO led to the most marginal decline in $T_{g}$ as the oil content increased. Certainly, from DSC, the highest recorded $T_{g}$ among all types of oils was observed for EAS at the highest bio content (30%).

Samper et al. [59] combined commercial ESO and epoxidized linseed oil (ELO) with varying concentrations (20, 40, 60, 80, and 100 wt%) to develop a fully bio-based epoxy system, except for the curing agent, which was prepared by combining phthalic anhydride (PA) and MA. The inclusion of ELO led to an improvement in the flexural properties of the ESO-based epoxy system, attributable to the greater epoxy group content per molecule in the ELO structure compared to ESO. As shown in Table 11, the increase in ESO negatively affected either the strength and modulus in flexure due to the higher rigidity of ELO given by its inherent higher crosslinking density. That reflected also in a higher $T_{g}$ value for higher ELO percentage. Conversely, the impact strength exhibited an almost linear correlation with the ESO content, culminating in a maximum value of 6.9 kJ/m${}^{2}$ for a 100% ESO composition.

Pin et al. [61] assessed the influence of various crosslinkers, initially MHHPA and then 4,4′-tetracarboxylic dianhydride (BTDA), on epoxy resins derived from ELO. In the case of BTDA, the ELO percentage went up to 70%, and only 60% instead for MHHPA. The former, indeed, allowed an increase in the crosslinking density leading to the possibility of incorporating more bio content, not invalidating mechanical performance.

Sahoo et al. [62] pushed up the realization of sisal composites based on the ELO-epoxy thermoset, with 10, 20 or 30 phr of oil content. As previously observed, a better natural fiber/matrix adhesion was expected with respect to that with traditional commercial epoxy (LY556). Hence, the mechanical and adhesive properties of composites experienced an improvement from the incorporation of ELO up to 20 phr (further bio content increase led to an over-plasticization).

Lerma-Canto et al. [64] utilized epoxidized *hempseed oil* (EHO) as a precursor to develop bio-based epoxy systems, with an emphasis on investigating their thermomechanical properties. This was achieved through the incorporation of MNA and maleinized hemp oil (MHO) as crosslinkers (with proportion ranging from 0 to 100%), resulting in the attainment of a completely bio-based epoxy configuration. The increase in MHO content flexibilized the final thermoset, leading to a reduction in flexural strength, modulus, and $T_{g}$ value as outlined in Table 12. Moreover, Charpy impact tests demonstrated strengths of 17.6 kJ/m${}^{2}$ and 17 kJ/m${}^{2}$ for configurations utilizing 75 and 50% of MHO, respectively [104]. It is worth noting that further increase in the MHO percentage resulted in the failure of the tested samples. Therefore, it would be of interest to conduct additional tests to assess the nature of these failures and to determine the maximum potential increase in impact strength achievable through the incorporation of hemp oil. The obtained performance is not comparable to that guaranteed by the fully EHO-based epoxy system in [59]. Manthey et al. [60] conducted a comparative study on the influence of incorporating ESO and EHO at varying concentrations (0, 10, 20, 30, and 40%) in epoxy (R246TX) blends, which were subsequently cured using isophorone diamine and triethylenetetramine. The samples composed of commercial epoxy and EHO exhibited superior flexural performance. However, it was observed that the maximum bio content (40%) exhibited inferior performance in flexural properties, both in comparison to the hemp-based epoxy thermoset outlined in Lerma-Canto et al.’s study [64] and the conventional epoxy thermoset (refer to Table 12). Beyond 30% bio content, there was a significant reduction in the crosslinking density, resulting in a notable decline in the mechanical properties of the epoxy. Consequently, it is not feasible to consider these vegetable oils and curing agents as consistent replacements for DGEBA. Furthermore, the higher EEW of ESO led to a $T_{g}$ value that exceeded that of EHO.

Shibata et al. investigated the effect of incorporating *glycerol* into epoxy systems in [38,71]. In particular, the latter exploited glycerol polyglycidyl ether (GPE) and polyglycerol polyglycidyl ether (PGPE) with ϵ-poly(L-lysine) (PL) as crosslinker, with 1:1 or 2:1 as the epoxy/amine ratio, for creating bio-based epoxies. The highest recorded $T_{g}$ value (47 °C) from DSC was exhibited by the latter with an epoxy/amine ratio 1:1, due to the greater crosslinking among the tested epoxies configurations. In [38], GPE and sorbitol polyglycidyl ether (SPE) were used to synthesize a new epoxy resin cured with TA. As already observed in [36], here, the enhancing effect in $T_{g}$ value through the increasing of TA as a crosslinker was stressed, with an additional improvement for SPE/TA due to the higher functionality of SPE compared to GPE. In terms of mechanical properties, it is important to observe that the GPE/TA thermoset showed a lower tensile strength but a greater modulus (2.4 vs. 1.7 GPa) than GPE/TA, where the latter property is comparable to that of available commercial bio-epoxies analyzed in Table 2. Barua et al. [70] utilized glycerol for the synthesis of hyperbranched epoxy resins (HBGEs) as promising coating epoxy systems due to the presented good adhesive strength and toughness. Unfortunately, the best configuration in terms of tensile strength value (48.6 MPa) corresponded to the lowest amount of glycerol (5%), attributable to the highest crosslinking density, while as the glycerol content increased up to 25%, the value halved. It is worth noting that even with a limited level of bio-incorporation, the achieved mechanical performance was still considerably lower compared to that of a conventional epoxy, like LY556 + HY917. Nevertheless, the addition of a bio-source had a positive impact on the elongation at the break of the epoxy, surpassing the value of its DGEBA-based counterpart due to its more brittle behavior.

Finally, it is important to stress that the analyzed edible oils can also be incorporated in epoxy systems simply as hardeners [63,64,65]. While this strategy certainly prioritizes the preservation of the mechanical performance of the final thermoset, it may not be as groundbreaking or innovative from a sustainability perspective as the direct incorporation of oils as epoxy monomers.

In this section, natural edible oils are just examined as potential sources for bio-based epoxy thermosets. Now, the analysis is extended to incorporate non-edible oils for their potential application in epoxy systems. In this context, it is worth describing the *castor oil* use, which is an abundant renewable resource. Ricinoleic acid (12-hydroxy-cis-9-octadecenoic acid), a hydroxyl fatty acid, constitutes its major component.

There is limited literature available on the use of non-drying oils, mainly because of the difficulty in achieving thermomechanical performance levels comparable to DGEBA, due to their aliphatic structure. For example, Park et al. [72] investigated the effect of epoxidized castor oil (ECO) blended with a DGEBA-based epoxy, with a latent thermal catalyst (BPH) used as a curing agent. The epoxidation or hydroxylation has to be necessarily performed to make non-edible oils more reactive [105], similar to what is observed for edible oils. The addition of ECO in the epoxy system (0–40% bio-content) reduced the rigidity of the epoxy network, which in turn also determined a lowering of $T_{g}$ as the bio content increased. It is noteworthy to mention that the inclusion of oil resulted in an improvement in flexibility, in contrast to the outcomes observed for lignin and TA. In particular, an increase in ECO content up to 30 wt% led to an enhancement in the flexural strength (from 88 to 117 MPa), while the flexural modulus remained unaffected by the bio-content. In [73], up to 50% of ECO was blended with a commercial DGEBA-based epoxy (Araldite GY 250) cured with ARADUR HY915. Increasing bio-incorporation still led to a stiffening of the epoxy, attributed to a reduction in crosslinking and, consequently, a decrease in the $T_{g}$. Table 13 illustrates the decrease in tensile properties with the increasing content of ECO.

Kadam et al. [37] synthesized a fully bio-based epoxy resin from TA and epoxidized *karanja oil* (EKJL), which is composed of oleic acid as the major fatty acid. The tensile mechanical strength demonstrated improvement upon substituting TA with EKJL, increasing from 5 to 11 MPa. However, it did not reach comparable levels to those of petroleum-based epoxy resins but surpassed them in terms of thermal stability.

Omonov et al. [74] formulated epoxy thermosets based on epoxidized *canola oil* (ECNO) [106] and cured them with PA, with respective molar ratios of 1:1, 1:1.5, and 1:2 mol/mol. The study focused on the coupling effect of the curing temperature and ECNO/PA ratio. The $T_{g}$ value is significantly sensitive to the amount of PA; indeed, the temperature linearly increased with it. On the other hand, $T_{g}$ did not effectively vary with the curing temperature, which was set at 155, 170, 185, and 200 °C. The authors indicated that these thermosets could have practical uses in ligno-cellulosic reinforced composites, which are known to be particularly sensitive to high temperatures. This is made possible by carefully managing the type and quantity of crosslinkers, which accelerates the curing process and improves the mechanical properties.

## 3. Bio-Epoxy Composites

In the following sections, the discussion starts from bio-epoxy composites made of synthetic fibers (carbon and glass). Since few are the works that handle these fibers in combination with biological resins, the focus shifts toward hybrid reinforcements: in particular, carbon/flax, carbon/areca, carbon/bamboo, basalt/flax, basalt/areca, and basalt/bamboo are analyzed. The investigation then moves to purely natural composites, where the properties of the following laminates – hemp, jute, sisal, flax, ixtyle, henequen, and bamboo—are discussed.

### 3.1. Synthetic and Hybrid Composites

Talking about synthetic fibers, we cannot avoid referring to carbon and glass.Therefore, it is worth mentioning a couple of recent papers dealing with the mechanical properties of carbon fiber-reinforced composites produced with epoxy and bio-based epoxy resins, comparing their behavior [68,69]. The commercial bio-based epoxy considered by the authors was IB2 (31% bio-content), whose properties are already discussed in Table 1 and Table 2. The obtained composites were subjected to tensile, compression, bending, and low-velocity impact (LVI) tests to characterize them and assess their applicability. To facilitate the reader in comparing the results presented below, Table 14 and Table 15 summarize the basic parameters of the investigated fabrics and the most important properties of some conventional epoxy composites used as the benchmark, respectively.

From the data reported in [68,69], no remarkable differences in the mechanical response emerged from the comparison between carbon/epoxy (see Table 15) and carbon/bio-epoxy composites: in particular, a reduction between 5 and 10% in the tensile and flexural properties was observed. To complete the material investigation, LVI tests at 15 and 30 J were performed on 2 mm thick specimens. Also in this case, the different types of resin did not seem to affect the values of peak force and absorbed energy; in both cases, at 15 J, a rebound of the dart of about 1 mm was observed, and perforation was achieved at 30 J. This demonstrated that a bio-content of up to 31% did not affect the structural integrity of the laminates and, therefore, constitutes a possible solution to replace traditional composites.

A similar comparison can be made for glass fiber composites: Ferdosian et al. [28] investigated how the mechanical properties changed, taking into account a variation of the DHL bio-content between 25 and 100%. The results showed that the specimens with a DHL content up to 50% can be considered possible substitutes for conventional glass/epoxy composites. Compared to the values in Table 15, a reduction of 14 and 24% was observed in the tensile and flexural strength, respectively. A higher percentage of DHL (i.e., greater than 50%) led to a reduction in the mechanical properties due to poor fiber–matrix bonding, which can be seen in the SEM image below (Figure 9) for 100% DHL epoxy resin.

To reduce the composites’ dependence on fossil and petrochemical resources, hybrid reinforcements in combination with bio-based thermoset matrices can be used. From the available literature, three recent papers [16,17,18] dealing with the SR56 + SD bio-based epoxy matrix were selected. As for the reinforcements, the authors analyzed the combination of synthetic and mineral fibers—such as carbon (C), and basalt (B) used as outer layers—with natural counterparts, such as flax (F) [16], areca (A) [17], and bamboo (Bb) [18], as cores. All the hybrid composites were produced by hand lay-up and the subsequent compression molding technique, and were investigated from both the mechanical and thermal points of view. Tensile and Izod impact trends, along with thermal and water absorption considerations, are hereinafter discussed. Comparing the tensile performances, no evident differences were observed using flax or areca as the core layers, while the presence of bamboo seemed to reduce the composites’ capabilities more. On the contrary, the substitution of the carbon external layers with basalt ones led to a significant decrease in the tensile properties by at least 40%. These reductions were related to the higher strength of carbon but also to the better interaction between its functional group and the epoxy matrix. An opposite trend was observed for the impact strength: in all cases, hybrid composites containing basalt fibers provided better results due to their higher energy absorption capability. In particular, this ability is strictly related to the following fracture modes: fiber pull-out, fiber breakage, plastic deformation, and fiber–matrix debonding [18]. Compared with the results in Table 15, bio-based hybrid composites showed a mechanical properties reduction of approximately 25%.

According to ASTM D570, the water absorption test was performed to evaluate the suitability of these hybrid bio-composites for long-term service applications. This aspect affected both the polymer and fiber, resulting in the deterioration of the properties. It is evident that conventional epoxy laminates are more resistant to water absorption with respect to bio-based epoxies because of the hydrophilic character of fibers and the fiber–matrix interface adhesion [16]. Moreover, with respect to pure polymers, composites absorb more water since they can freely enter through the fibers, modifying the polymer–fiber interface [18].

To complete the discussion on hybrid bio-based composites, some considerations about the thermal stability were added through the analysis of the derivative thermogram (DTG) curves. It was found that bio-based epoxy composites had thermal stability for temperatures up to 290 °C, and therefore they can be potentially adopted in applications workable till this temperature. Finally, only Yorseng et al. [18] discussed $T_{g}$. In particular, the addition of carbon/bamboo and basalt/bamboo fabrics increased the $T_{g}$ value of the neat bio-epoxy resin from 79 to 93 °C, and the same values were found by the authors also for the same reinforcement embedded in synthetic epoxy resin. The accelerated weathering test was performed to analyze the long-term performance of composites at different environmental conditions, according to ASTM G155-13 [107], and it revealed only a marginal drop in properties. Thus, these bio-based epoxy composites can be safely used to replace synthetic epoxy hybrid composites since they can be applied in all weather conditions as confirmed by the above results.

### 3.2. Purely Natural Fiber Composites

Composite materials consisting of natural fiber reinforcements and bio-based epoxy matrices represent the best configuration in terms of sustainability. Despite the environmental advantages of using natural fibers, some drawbacks can significantly affect the performance of the final products. For example, natural fibers tend to absorb large amounts of moisture, leading to a reduction in fiber–matrix adhesion, which is a fundamental key for the composites mechanical properties. Moreover, unlike synthetic fibers, the performance of natural fibers varies according to harvesting and extraction techniques, ground and climatic conditions, and chemical treatments. Among the most commonly used natural fibers, hemp is one of the most economical and readily available bast fiber in Europe. It is characterized by high specific mechanical properties and high cellulose content, which makes it usable as reinforcement in polymer matrix composites [108]. For this reason, the discussion begins by comparing the mechanical and impact properties of hemp fiber bio-composite laminates as a function of the percentage of bio-content in the epoxy matrix. Scarponi et al. [108] considered a bio-content percentage of 21%, which is the cut-off point for our study. They dealt with LVI and bending tests on 5 mm thick samples having a fiber volume fraction of 42%. The results highlighted that hemp/bio-epoxy composites had very good impact properties; the peak force, the maximum displacement, and the absorbed energy were evaluated from 5 to 40 J, which corresponds to perforation. In addition, the residual bending properties were also discussed. No differences in the LVI load-displacement curves between bio-epoxy and conventional epoxy composites were observed. In particular, when the load reached its peak and dropped suddenly, an irregular plateau was noted, as severe internal damage developed. For impacts up to 20 J, the impactor rebounded, and no visible damage was observed on the front of the specimens, but cracks in the matrix and splitting of the surface were detected on the back. Bio-epoxy composites showed slightly more localized damage, and no clear gaps between the hemp fibers and matrix were registered, differently from the synthetic counterpart. To better understand the involved failure mechanisms, Figure 10 was added. It shows the damage mode of hemp/bio-epoxy laminates through SEM micrographs, revealing an effective impregnation between fiber and resin, promoting a good interface. Red arrows indicate the direction of kinks formation, as the effect of solid fiber-matrix bonding.

Flexural properties, measured before and after impact, showed that the reduction in strength and stiffness was higher in conventional epoxy composites at impact energies up to 20 J. In summary, laminates made with 21% bio-epoxy resin offered similar, if not better, properties to conventional epoxy resins in terms of impact resistance, due to improved fiber–matrix adhesion. In increasing the bio-content percentage, Colomer-Romero et al. [109] investigated the tensile and flexural properties of hemp/bio-epoxy composites using the bio-based epoxy system SP100 with a bio-content of 37%. The results showed that the properties were reduced by 20%; therefore, they can replace conventional hemp/epoxy laminates in certain applications, e.g., small wind blades.

Since in many cases the mechanical performance results of a single natural fiber are not so good, the hybridization technique could be a way to solve this problem. Combining two different types of natural fibers can help to improve the overall performance of the produced composites. For example, by combining hemp (H) and jute (J) fibers, we can take advantage of the toughness of hemp with the elongation typical of jute. This specific hybrid configuration was investigated by Vinod et al. [110], who compared the performance results of pure materials (see Table 16) with different hybrid stacking sequences using a SR56 + SD bio-epoxy system. Tensile, flexural, and Izod impact tests were analyzed. Based on the tensile and bending results, the strength of hemp/jute/hemp (H/J/H) composites was increased by 16% compared to pure hemp, and it was reduced by 19% when hemp and jute were inverted: J/H/J. At the same time, J/H/J offered better mechanical properties than pure jute with a 16% increase in strength, while there were no evident differences compared to pure hemp. An opposite trend can be observed for impact strength: in this case, pure jute composites showed the best performance due to the high elongation, which facilitates the energy absorption capacity in case of an impact. The high stiffness typical of hemp fibers led to sudden failure on impact, which explained the degradation of impact performance observed in hybrid and pure hemp configurations. This suggested that the use of hemp fibers as the outer layers increased the tensile strength and counteracted the bending loads, while jute as a core layer prevented the failure of the composite. Although in this case hybridization offered some advantages over pure fiber composites, the combination of hemp with sisal did not lead to the same results. This was explained by Thiagamani et al. [111] and Senthilkumar et al. [112], who studied the effects of different hemp and sisal hybrid stacking sequences, revealing that pure composites provided comparable and sometimes better results with respect to hybrid solutions. All investigated hybrid configurations showed slightly worse or similar results compared to the pure composites, indicating poor compatibility between the two types of fibers: no significant differences were found by changing the stacking order. With regard to possible applications, a comparison of the most important properties before and after weathering tests was carried out [112]. In particular, marginal differences were registered in the tensile strength of pure composites before and after the test, while an unfavorable effect was evidenced on the bending properties: pure, but especially hybrid composites showed lower strength and modulus compared to unweathered specimens. This can be attributed to the accelerated weathering, which led to the penetration of UV rays into the epoxy matrix and caused degradation and fibers debonding.

Besides the reinforcements discussed up to now, one of the most widespread and used fiber in the production of natural composites is flax. Yashas Gowda et al. [16] focused on the mechanical properties of composites made up of flax and the SR56 + SD system, along with hybrid solutions already discussed in the previous subsection. Comparing these results with the ones by Vinod et al. [110], and looking at Table 16, it emerged that the use of flax, jute or hemp as natural reinforcement did not affect in an evident manner the tensile and Izod impact properties. However, the mechanical properties of natural fiber composites were affected by environmental changes as suggested by Moudood et al. [113], who assessed the performance of flax/SR56 + SD bio-epoxy composites under different thermal conditions. A small reduction in the mechanical properties was observed if the composites were exposed to a warm, humid or freeze/thaw cycling environments. On the contrary, water immersion reduced drastically (about 46%) the flexural properties and the resistance to buckling. This happened because water caused the degradation of the fiber–matrix interface, which led to a premature composite failure. Therefore, from the obtained results, it can be concluded that flax/bio-epoxy samples can operate in most environmental conditions, except for water immersion, which caused important degradation in the mechanical properties. However, even if the laminates were immersed in water for a long time, the authors stated that the flexural properties and also the tensile modulus can be completely restored, drying the composites, while nothing can be done for tensile strength.

In addition to these commonly used natural fibers, also other plant-derived fibers can be used to manufacture bio-based epoxy composites. Torres-Arellano et al. [114] compared the mechanical and thermal properties of two commercial bio-based epoxy matrices reinforced with jute and two agave-derived fibers: ixtle and henequen. The bio-based epoxy matrices considered in the study were (1) SRGreenpoxy56 + SZ8525 (SR56 + SZ) with 40–50% bio-content, and (2) EVO with 40% bio-content. From the tensile tests, it appeared that laminates made with the first bio-epoxy system generally gave better results than EVO composites, indicating better compatibility. Although it is not possible to define a direct relationship between the flexural properties and the bio-content, it can be stated that, also in this case, the best performance was obtained by jute/SR56 + SZ composites. As far as the resin and hardener selection is concerned, Table 17 shows that the comparison of two composites with the same fiber and the same resin but with a different hardener led to distinctly different mechanical results. The SZ hardener gave better results in terms of strength but above all in terms of the modulus, considerably stiffening the composite.

All the papers analyzed so far considered resin systems with a bio-content from 21 to 50%, yet there were a few papers in which the authors attempted to formulate a fully biological epoxy resin. An example is the work by Bagheri et al. [21], where the authors investigated composites reinforced with bamboo fibers and 100EKL/NT1515, a fully bio-based thermoset resin, whose properties were already discussed in Section 2.2.1. As the lignin in plants acts as a natural adhesive, an excellent interaction between lignin and bamboo fibers was observed, leading to slightly superior flexural properties with respect to traditional bamboo/epoxy composites. This aspect provided also an improvement in the DMA results, revealing that fully bio-based composites achieved better thermal stability and were capable of operating at higher temperature. Another attempt to introduce fully bio-based materials was represented by cellulose long filament (CLF)-reinforced vanillin epoxy (VEP) composites. They were manufactured by Adil et al. [115] through vacuum-assisted resin transfer and compression molding techniques. Due to the strong interfacial bonding between the CLFs and the bio-based matrix, the flexural, thermal, and water absorption properties were very good. Therefore, they represent a very recent example of lightweight high-performance fully green composites.

## 4. Proposed Applications

The thermomechanical and adhesion properties, along with the low cost of bio-based epoxy resins, suggest extensive practical applications as coatings and adhesives as discussed in previous sections. Furthermore, these applications demonstrate an enhancement in the corrosion resistance properties of conventional epoxy systems when using bio-based raw materials. The subsequent sections begin with proposed practical applications of neat bio-epoxy resins before moving on to those within the composites field. In this context, a few ambitious applications, such as the use of bio-epoxy composites in structural applications for aircraft or racing cars, are limited by the still non-competitive mechanical performance of these bio-materials compared to traditional epoxies. However, they are effectively employed for coating or 3D-printing applications.

### 4.1. Neat Bio-Epoxies

Despite the efficiency of epoxy coatings, long exposure to environmental parameters might compromise their corrosion resistance. Due to the presence of phenolic compounds and functional groups, lignin, vanillin, and TA have the potential to inhibit oxidation. In [116], the durability of epoxy as a coating system for steel components was improved through the addition of acetylated lignin, leading to a significant enhancement compared to traditional coal tar epoxy, without compromising the adhesive properties. Chang et al. [117] investigated the synergic enhancing effect of TA, DA-LIM, and nano-ZrO${}_{2}$ on the corrosion resistance of epoxy as coating protection for rusted steel. The rust conversion action of TA and the enhancement of the adhesion properties with steel substrates were further fostered by the presence of DA-LIM, preventing the coating/substrate interface from being touched by corrosion agents and avoiding TA leakage. Protection from UV radiation and photodegradation can also be successfully achieved through epoxidized vegetable oils as attested by Ammar et al. [118], who discussed the benefits of ESO in reducing corrosion in steel coatings. Branciforti et al. [119] practically used ELO and ESO for stereolithography, ensuring UV absorption as alternatives to acrylate monomers. These latter are a source of toxicity, which was observed to persist even after UV light treatment following the curing of 3D objects. For this reason, epoxidized vegetable oils were used as ecological alternative solutions. Coating application for mild steel and aluminum substrates was proposed by Kathalewar and Sabnis [120] for a cardanol-based epoxy resin. Bio-content from 40 to 50% ensured improved corrosion resistance compared to a counterpart DGEBA-epoxy system.

### 4.2. Bio-Fillers and Epoxy Composites

Comprehensively addressing sustainability concerns and achieving a satisfactory level of mechanical performance is imperative to enable practical applications of composites. To satisfy these requirements, natural and biodegradable waste feedstocks—such as eggshell [121], seashell [122], plant char [123], coffee grounds [124], or even chicken feathers [125]—can be used as fillers in the matrix. The above-mentioned literature reveals that the addition of these substances enhanced the mechanical and thermal properties together with the amount of renewable content.

Apart from fillers, some research has analyzed the possible bio-composites applications. An example is the work by Ghoushji et al. [126], which investigated the crashworthiness properties of ramie/bio-epoxy composite square tubes with different lengths, and assessed their suitability as potential energy absorbing components. From static axial compression tests, the best properties in terms of average load and specific energy absorption (SEA) were observed for short tubes, and a significant improvement in SEA was detected as the number of layers increased. Since the geometry of the tubular structure also affected the performance results, more geometries need to be investigated to obtain a more complete idea of the suitability of ramie/bio-epoxy tubes for crash applications.

Another field in which bio-epoxy materials can be applied is the aviation sector. In this context, Ramon et al. [127] provided an overview of different bio-based epoxy systems (derived from natural oil, furan, lignin, and rosin) and compared their mechanical and thermal performance results with the petroleum-based systems already used in this sector. The results highlighted that these innovative resin systems can be considered favorable candidates to replace conventional epoxy resins for aircraft interiors, although some work still needs to be conducted. Another attempt in this regard was made by Yi et al. [128], whose described their efforts to develop an interior side panel for an aircraft and the body of an electric racing car using honeycomb sandwich composites and a 30% bio-content rosin-based epoxy resin.

Interesting applications were identified in connection with the use of vegetable oils such as ESO and karanja oil. In the first case, Pansumdaeng et al. [56] demonstrated the potential of fully bio-based ESO epoxy thermosets for the production of energy harvesting devices. In the second case, Kadam et al. [37] proposed an alternative application for the karanja oil-based epoxy thermoset. In addition to the examination of mechanical properties and biodegradability, a larvicidal effect against mosquitoes on paper coating was analyzed. Hence, the application of karanja oil-based bio-epoxy on a paper coating revealed a valid protection against mosquito larvae, presenting a compelling solution to control dengue fever transmission through these vectors.

Finally, also 3D printing involves bio-based materials: resorcinol epoxy acrylated (REA) resin combined with synthetic fibers—such as glass, nylon, and polyester fibers—was used to create objects with this technology. A complete characterization of these three composites was made by Desai and Jagtap [34], who declared that nylon- and polyester-reinforced composites exhibited the best properties to be applied in this field.

## 5. Outlook and Challenges

Bio-epoxy, as is known, has been developed from wood wastes to replace BPA and ECH. In this context, considering that the incorporated bio-components in epoxy can potentially be derived from any lignocellulosic material with a vast range of applications, and given the ever-increasing widespread use of natural fibers, current research is focused on optimizing bio-content. The goal is to tailor the properties necessary for the epoxy matrix and enhance compatibility with natural reinforcements in composite applications. In the pursuit of promoting eco-friendly epoxy systems, it should be emphasized that choosing bio-raw materials derived from waste is preferable to synthesized products, whose sustainability might be questionable. The use of virgin synthesized materials might easily allow the incorporation of a high bio-content, which can also be optimized as demonstrated in [129]. A multi-objective optimization approach was proposed to simultaneously maximize the $T_{g}$, flexural strength and modulus within the bio-content range of 20–35%. In that case, the well-known nature of synthesized products leads to a more accurate curing process and, consequently, efficient mechanical performance of the final resin. On the contrary, the uncertain nature and characterization of waste-derived products negatively affect the efficiency of the curing, resulting in compromised mechanical, thermal, and chemical properties of the resulting resin. In such cases, a conservative over-curing process is usually adopted. For these reasons, it is crucial to optimize the process to enhance the properties of the resin as well as possibly reduce the energy consumption, time, and cost, before taking into account the maximization of bio-content in the epoxy system. In this regard, Lascano et al. [130] provided an optimization of the curing and post-curing temperature for enhancing the thermomechanical properties and gel time of a commercial bio-epoxy resin (31 wt% bio content). The study revealed that a reduced gel time and improved toughness can be achieved under optimal moderate curing temperatures, followed by high post-curing temperatures. In [131], different curing agents were tested to optimize the crosslinking density of a magnolol-based epoxy, alongside the regulation of the rigidity of structural units in epoxy oligomers. In particular, moderate rigidity of epoxy oligomer structural units yielded outstanding mechanical properties in the cured resins. The optimized curing process resulted in epoxy with improved heat resistance, thermal stability, and optimal char residue with respect to first-time cured epoxies.

## 6. Conclusions

For more than two decades, due to the high toxicity of BPA and ECH required for the production of DGEBA and thus for the synthesis of epoxy resins, attempts have been made to solve this problem by replacing these two components with bio-based products. Therefore, the introduction of some amount of bio-based content either into the resin or in the curing agent is especially related to the need for improved sustainability. More specifically, the aim is to reduce resource depletion, and to limit the toxicity in the production and disposal of epoxy products.

It is noteworthy that this trend has gained considerable attention, along with the increasing interest for the application of natural fibers as reinforcements in composites, often proposed in hybrid solutions with synthetic fibers. Over time, also commercial bio-epoxy resins have been developed: however, ongoing research is increasingly focused on advancing the replacement of traditional epoxies. In this context, difficulties arise by the limited extent of oil-based content in epoxy formulations. This restriction aims to preserve the mechanical performance of resins and composites, as well as avoiding alterations in their thermal properties, particularly concerning the $T_{g}$ and degradation pattern. This suggests that the attention needs to focus on the analysis of the chemical building blocks, involved in the synthesis of bio-based epoxies. In particular, the bio-based content is primarily developed from lignin raw materials, although the originating botanical species are not always defined. Alternatively, lignin-derived products and food wastes are used. Among these are especially cardanol, vanillin, resorcinol, eugenol, or oils more directly related to food byproducts, such as castor oil, soybean, hempseed or linseed oil.

These bio-epoxies were adopted with several natural fibers (e.g., flax, areca, and bamboo), also stacked into hybrid configurations with carbon or basalt fibers; notwithstanding, it still remains open the competitiveness issue related to the traditional epoxies’ properties. In this respect, the completion of the bio-based character, together with the performance improvement, is also proposed with other natural bio-fillers. Nevertheless, it is reasonable to assert that the widespread application of bio-epoxies in the primary sectors, traditionally chosen for structural and highly crosslinked thermosets, is not yet imminent.

## Figures and Tables

**Figure 1 polymers-15-04733-f001:**
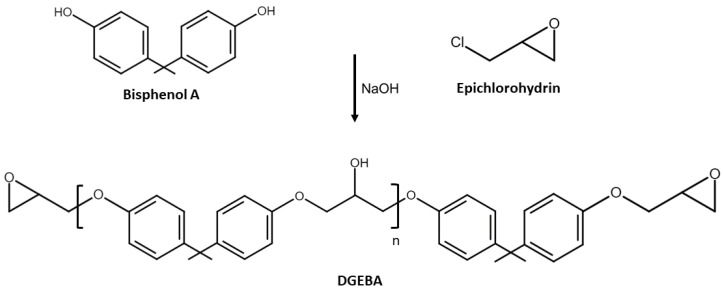
Synthesis of epoxy resin from BPA and ECH.

**Figure 2 polymers-15-04733-f002:**
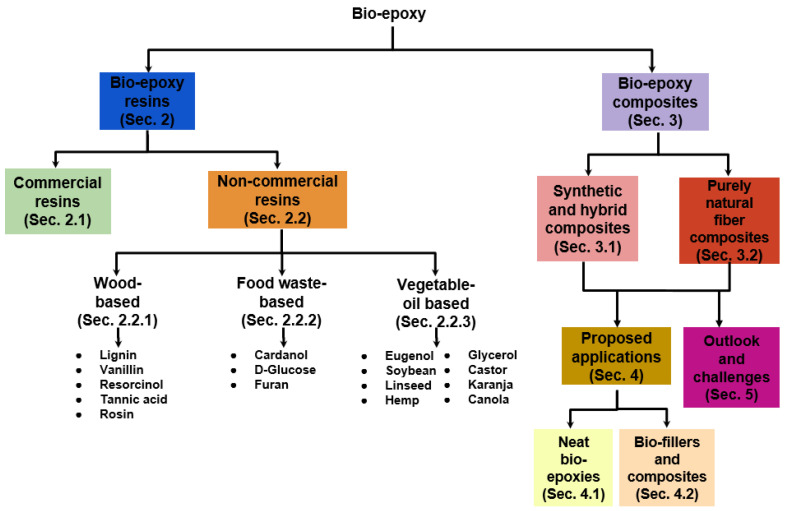
Block diagram of the framework.

**Figure 3 polymers-15-04733-f003:**
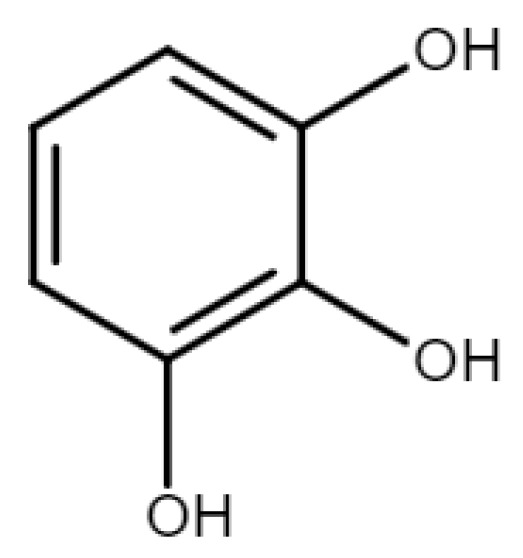
Pyrogallol C_6_H_3_(OH)_3_ chemical structure.

**Figure 4 polymers-15-04733-f004:**
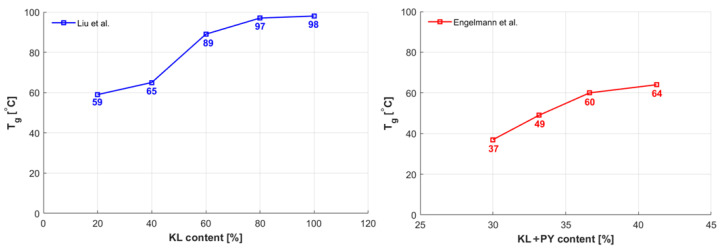
The $T_{g}$ of thermosets with different KL content by Engelmann et al. [22], and Liu et al. [24].

**Figure 5 polymers-15-04733-f005:**
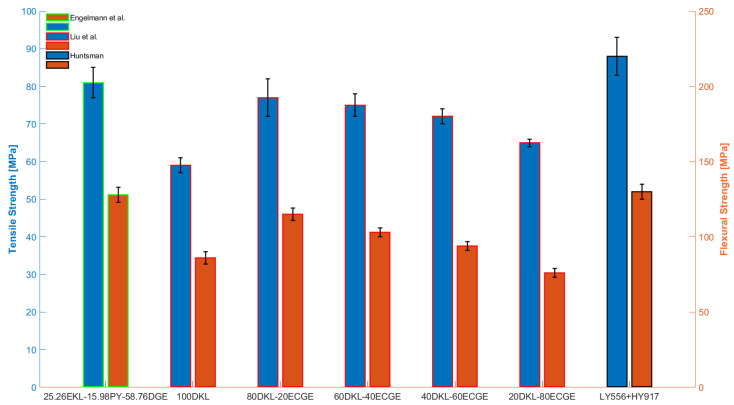
Tensile (blue) and flexural (orange) strengths of different content of KL-based epoxy systems, by Engelmann et al. [22] (green) and Liu et al. [24] (red), along with a DGEBA-based reference. i.e., LY556 + HY917 by Huntsman [82], Klybeckstrasse, Basel, Switzerland (black).

**Figure 6 polymers-15-04733-f006:**
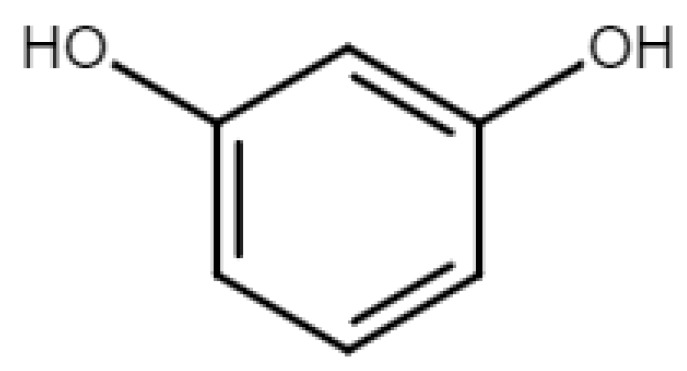
Resorcinol C_6_H_6_O_2_ chemical structure.

**Figure 7 polymers-15-04733-f007:**
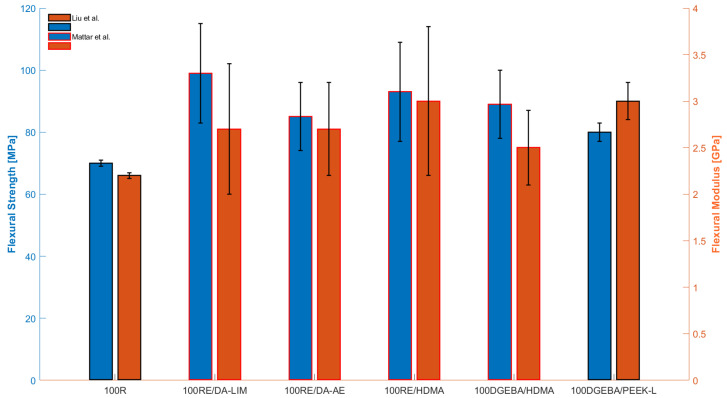
Flexural strength (blue) and modulus (orange) of fully and partially bio-based (R or RE) epoxy systems and DGEBA references by Liu et al. [41] and Mattar et al. [32,33].

**Figure 8 polymers-15-04733-f008:**
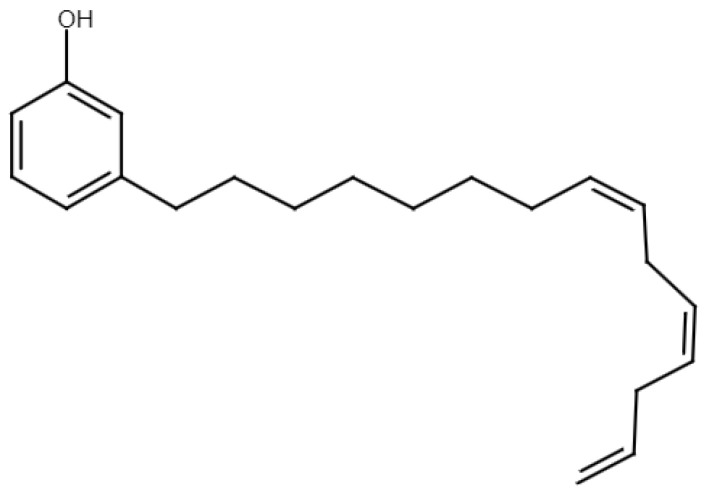
Cardanol chemical structure.

**Figure 9 polymers-15-04733-f009:**
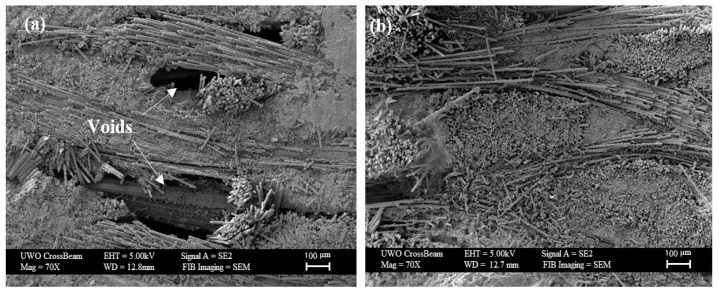
Scanning electron micrographs of glass fiber-reinforced composites: (**a**) 100% DHL epoxy resin, and (**b**) synthetic epoxy resin [28].

**Figure 10 polymers-15-04733-f010:**
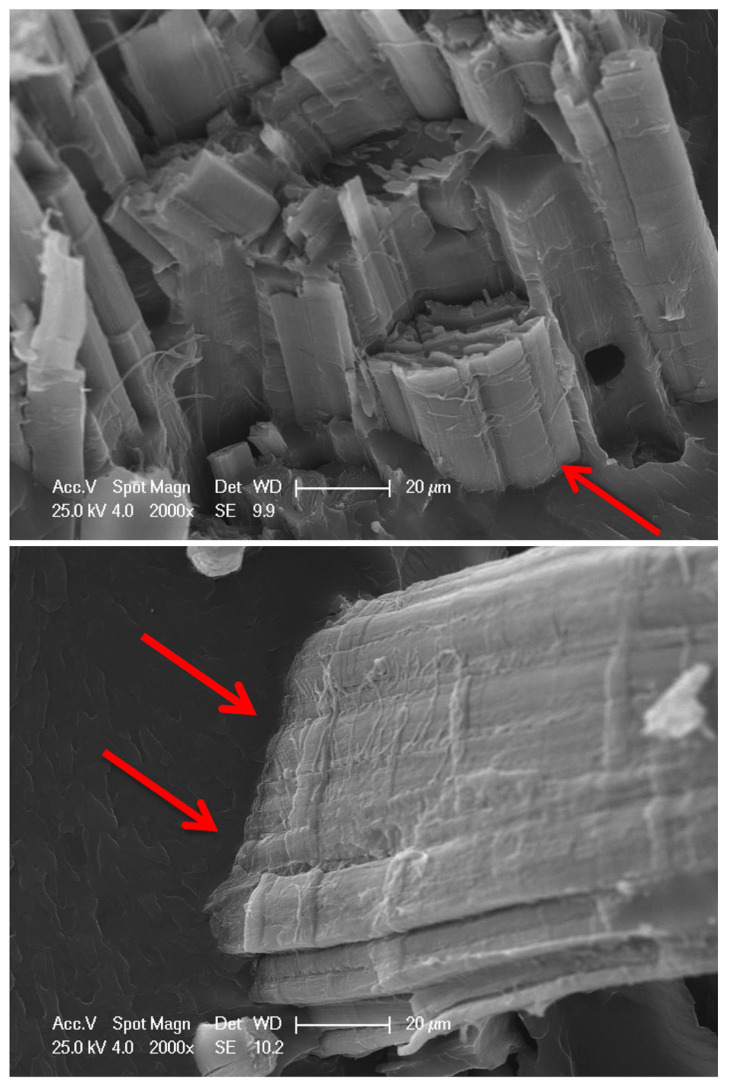
SEM micrographs showing fracture surfaces of hemp/bio-epoxy laminates [108].

**Table 1 polymers-15-04733-t001:** Bio-based resin systems categorized by bio-feedstock origin, its percentage content, and curing agent, together with DGEBA-based counterpart (N.D., not declared).

Name	Epoxy System	Bio-Feedstock Nature	Bio Content	Amine Type	Ref.
AMB	AMPRO BIO/ Slow Hardener (Gurit AG, Zurich, Switzerland)	Cashew nut shell liquid	40–60%	cycloaliphatic, aliphatic and aromatic	[75]
EVO	SR Surf Clear EVO/ SD EVO Fast (Sicomin, Châteauneuf-les-Martigues, France)	Vegetable oil	40%	N.D.	[76]
SR56 + SD	SRGreenpoxy56/ SD7561 (Sicomin)	Plant origin	35–41%	aliphatic	[77]
SP100	SuperSap 100/ 1000 Hardener (Entropy, San Francisco Bay Area, CA, USA)	Waste pine and vegetable oils	37%	cycloaliphatic and aliphatic	[78]
F2501	FORMULITE 2501A/ FORMULITE 2401B (Cardolite Corp., Bristol, PA, USA)	Food waste origin: cardanol	34%	aliphatic and cycloaliphatic	[79]
BP36	BioPoxy 36/ Clear Hardener (Ecopoxy, Winnipeg, MB, Canada)	Soybean, cashew nut oil and recycled egg shells	32%	aliphatic and aromatic	[80]
IB2	IB2/ Amine Hardener (Easy Composites, Stoke-on-Trent, UK)	Plant origin: glycerol	31%	N.D.	[67]
INF810	Infugreen810/ SD8824 (Sicomin)	Plant origin	29%	aliphatic and aromatic	[81]
LY556 + HY917	Araldite LY556/ Aradur 917 (Huntsman Corp., Freeport, TX, USA)	Petroleum based	0%	aliphatic	[82]

**Table 2 polymers-15-04733-t002:** Mechanical properties of available commercial bio-epoxy resins and hardeners systems and DGEBA-based counterpart.

Name	Tensile Strength [MPa]	Tensile Modulus [GPa]	Flexural Strength [MPa]	Flexural Modulus [GPa]
AMB	36	1.9	62	1.8
EVO	68	3.4	117	3.2
SR56 + SD	70	3.1	114	3.2
SP100	57	2.6	77	2.3
F2501	69	3.1	113	2.8
BP36	58	2.8	97	2.9
IB2	65	2.8	107	2.8
INF810	62	3.1	106	9.9
LY556 + HY917	88	3.2	130	-

**Table 3 polymers-15-04733-t003:** Characterization of lignin-based epoxy systems according to lignin concentration and selected hardener system.

Epoxy System	Lignin Type	Bio Content	Hardener Nature	Ref.
EKL/NT1515	Kraft	100%	cashew nut shell based (Cardolite Co., Bristol, PA, USA)	[21]
EKL-DGEBA/PY	Kraft	20–50%	gallic acid based (Sigma-Aldrich, Burlington, MA, USA)	[22]
EKL-DGEBA/LE20	Kraft	15–30%	petroleum based (ABCOL, São Caetano do Sul, SP, Brazil)	[23]
DKL-DGEBA/MeTHPA	Kraft	20–80%	phthalic anhydride based (Jiaxing Lianxin Chemical New Materials Co., Zhejiang, China)	[24]
DOL-DGEBA/DMM or DKL-DGEBA/DDM	Kraft & Organosolv	25–100%	aniline based (Sigma Aldrich)	[25]
GOL-DGEBA/IPDA	Organosolv	0–33%	nitrile based (TCI, Tokyo, Japan)	[26]
DHL-DGEBA/DDM	Hydrolysis	25–100%	aniline based (Sigma Aldrich)	[28]

**Table 4 polymers-15-04733-t004:** Thermal properties of DHL-based epoxy thermosets and DGEBA-based counterpart [28]. * Araldite^TM^ GZ 540 × 90 epoxy.

Thermoset Sample	$T_{d5\%}$ [°C]	$T_{\max}$ [°C]
100DHL/DDM	213	368
75DHL/DDM	232	372
50DHL/DDM	240	377
25DHL/DMM	278	384
100DGEBA */DDM	360	405

**Table 5 polymers-15-04733-t005:** Thermomechanical properties of reprocessed vanillin-based epoxy vitrimers and DGEBA-based counterpart [29]. * E51 epoxy.

Thermoset Sample	Tensile Strength at Break [MPa]	Tensile Modulus [GPa]	$T_{d5\%}$ [°C]	$T_{g}$ [°C]
100Van-Ep/IPDA	65 ± 5	2.3 ± 0.2	222	121
100Van-Ep/IPDA-1	61 ± 4	2.4 ± 0.1	-	-
100Van-Ep/IPDA-2	63 ± 4	2.6 ± 0.2	-	-
100Van-Ep/IPDA-3	66 ± 3	2.6 ± 0.1	-	-
100DGEBA */IPDA	76 ± 4	2.5 ± 0.1	227	145

**Table 6 polymers-15-04733-t006:** Thermomechanical properties of vanillin-based epoxy vitrimers and DGEBA-based counterpart [30]. * E51 epoxy.

Thermoset Sample	Tensile Strength at Break [MPa]	Tensile Modulus [GPa]	Elongation at Break [%]	$T_{g}$ [°C]
100VE/DETA	53 ± 6	0.9 ± 0.1	9	58
100VE/IPDA	62 ± 7	2.8 ± 0.2	6	132
100VE/IPDA-D230	34	1.5 ± 0.1	8	84
100VE/DDM	48 ± 5	2.9 ± 0.3	2	143
100DGEBA */IPDA	76 ± 4	2.5 ± 0.1	5	145

**Table 7 polymers-15-04733-t007:** Thermomechanical properties of RE-based epoxy thermosets and DGEBA-based counterpart [32].

Thermoset Sample	Flexural Strength [MPa]	Flexural Modulus [GPa]	Elongation at Break [%]	$T_{g}$ [°C]	$T_{\max}$ [°C]
100RE/DA-LIM	99 ± 16	2.7 ± 0.7	8 ± 3	94	320
100RE/DA-AE	85 ± 11	2.2 ± 0.5	15 ± 2	97	328
100RE/HMDA	93 ± 16	3.0 ± 0.8	7 ± 2	110	353
100DGEBA/HMDA	89 ± 11	2.5 ± 0.4	6 ± 2	121	362

**Table 8 polymers-15-04733-t008:** Thermomechanical properties of partially and fully bio TAE-based epoxy thermosets and DGEBA-based counterpart [39].

Thermoset Sample	Tensile Strength [MPa]	Elongation at Break [%]	Impact Energy [kJ/m${}^{2}$]	Adhesive Strength (Wood–Wood) [MPa]	$T_{g}$ [°C]
100TAE/BA	18 ± 1	16 ± 1	19	4545 ± 37	62
90TAE-10DGEBA/BA	15 ± 2	39	17	2715 ± 29	64
75TAE-15DGEBA/BA	8 ± 2	55 ± 3	7	2040 ± 23	56
50TAE-50DGEBA/BA	5 ± 1	64 ± 1	5	881 ± 15	99
100DGEBA/BA	62 ± 4	8 ± 1	-	863 ± 5	72

**Table 9 polymers-15-04733-t009:** Thermomechanical properties of BOMF-based epoxy thermoset and DGEBA-based counterpart [50].

Thermoset Sample	Tensile Strength [MPa]	Tensile Modulus [MPa]	Shear Stress at Break [MPa]	Char Yield (700 °C) [%]	$T_{g}$ [°C]
100BOMF/MA	14 ± 1	382 ± 58	13 ± 5	26	34
100DGEBA/MA	51 ± 14	2187 ± 125	4 ± 2	16	56

**Table 10 polymers-15-04733-t010:** Thermomechanical properties of FF-based epoxy thermosets and DGEBA counterparts. * E51 epoxy.

Thermoset Sample	Flexural Strength [MPa]	Flexural Modulus [GPa]	Impact Strength [kJ/m${}^{2}$]	$T_{g}$ [°C]	Ref.
100FF/MeTHPA	38	2.2	3	99	[52]
93.3EUFU-EP/MHHPA	129	3.3	-	153	[51]
60FF-40DGEBA */MeTHPA	79	3.0	7	131	[52]
50FF-50DGEBA */MeTHPA	103	3.2	15	140	[52]
40FF-60DGEBA */MeTHPA	95	3.1	13	137	[52]
100DGEBA */MeTHPA	92	2.8	14	145	[52]
100DGEBA/MHHPA	140	3.0	-	144	[51]

**Table 11 polymers-15-04733-t011:** Thermomechanical properties of ESO- and ELO-based epoxy thermosets and DGEBA counterpart. (CA unspecified curing agent.) * Shell Epon 9500 epoxy.

Thermoset Sample	Tensile Peak Strength [MPa]	Tensile Modulus [GPa]	Flexural Strength [MPa]	Flexural Modulus [MPa]	$T_{g}$ [°C]	Ref.
100ELO/PA-MA	-	-	36	623	37	[59]
80ELO-20ESO/PA-MA	-	-	33	676	33	[59]
60ELO-40ESO/PA-MA	-	-	22	425	32	[59]
40ELO-60ESO/PA-MA	-	-	11	155	28	[59]
20ELO-80ESO/PA-MA	-	-	6	100	29	[59]
100ESO/PA-MA	-	-	1	11	27	[59]
30ESO-70DGEBA */CA	60	3.2	99	2910	62	[58]
30EAS-70DGEBA */CA	54	3.0	103	2979	65	[58]
30EMS-70DGEBA */CA	59	3.1	98	2841	55	[58]
20ESO-80DGEBA */CA	36	2.4	111	3090	67	[58]
20EAS-80DGEBA */CA	41	3.0	123	3359	69	[58]
20EMS-80DGEBA */CA	31	2.6	110	3083	63	[58]
100DGEBA */Epicure 9550	58	3.0	110	3021	75	[58]

**Table 12 polymers-15-04733-t012:** Thermomechanical properties of EHO-based epoxy thermosets and DGEBA counterpart. (CA Unspecified curing agent). * R246TX epoxy.

Thermoset Sample	Flexural Strength [MPa]	Flexural Modulus [MPa]	$T_{g}$ [°C]	Ref.
100EHO/100MNA	7	295	49	[64]
100EHO/75MNA-25MHO	6	100	34	[64]
100EHO/50MNA-50MHO	6	70	27	[64]
100EHO/25MNA-75MHO	1	13	20	[64]
100EHO/100MHO	1	12	7	[64]
40ESO-60DGEBA */CA	4	161	79	[60]
40EHO-60DGEBA */CA	6	311	80	[60]
30ESO-70DGEBA */CA	61	1490	88	[60]
30EHO-70DGEBA */CA	67	1701	90	[60]
20ESO-80DGEBA */CA	75	1895	91	[60]
20EHO-80DGEBA */CA	84	1935	97	[60]
100DGEBA */CA	108	2359	107	[60]

**Table 13 polymers-15-04733-t013:** Thermomechanical properties of ECO-based epoxy thermosets and DGEBA counterpart [73]. * Araldite GY 250 epoxy.

Thermoset Sample	Tensile Strength [MPa]	Tensile Modulus [GPa]	$T_{g}$ [°C]
50ECO-50DGEBA */HY915	18	0.901	39
30ECO-70DGEBA */HY915	42	2.728	47
20ECO-80DGEBA */HY915	54	2.131	72
100DGEBA*/HY915	70	3.343	97

**Table 14 polymers-15-04733-t014:** Main information of the discussed synthetic and hybrid/bio-epoxy composites: carbon (C), glass (G), basalt (B), flax (F), areca (A), bamboo (Bb), and N.D. (not declared).

Laminate	Woven Style	Weight (gsm)	Volume Fraction (%)	Ref.
C	Twill 2 × 2	210	51	[68]
G	Plain	Not available	N.D.	[28]
C/F	Plain	205 (C)–200 (F)	26.4	[16]
B/F		200 (B)–200 (F)	20.9	
C/A	Plain	205 (C)–200 (A)	34.9	[17]
B/A		200 (B)–200 (A)	33.2	
C/Bb	Plain	205 (C)–200 (Bb)	N.D.	[18]
B/Bb		200 (B)–200 (Bb)		

**Table 15 polymers-15-04733-t015:** Mechanical properties of synthetic and hybrid/epoxy composites.

Laminate	Tensile Strength [MPa]	Tensile Modulus [GPa]	Flexural strength [MPa]	Flexural Modulus [GPa]	Ref.
Carbon	716	57.8	658	41.7	[68]
Glass	214	17.5	266	13.0	[28]
Carbon/Flax	129	2.1	388	35.8	[16]
Basalt/Flax	89	1.2	171	14.2	[16]

**Table 16 polymers-15-04733-t016:** Mechanical properties of natural fiber reinforced SR56 + SD bio-epoxy composites.

Laminate	Volume Fraction [%]	Tensile Strength [MPa]	Tensile Modulus [GPa]	Impact Strength [kJ/m${}^{2}$]	Ref.
Hemp	23.7	56	1.9	2.5	[110]
Jute	21.5	47	0.9	5.9	[110]
Flax	29.4	47	1.0	2.6	[16]

**Table 17 polymers-15-04733-t017:** Comparison of the mechanical properties of jute/SR56 bio-epoxy composites with different hardener: SD and SZ.

Laminate	Volume Fraction [%]	Tensile Strength [MPa]	Tensile Modulus [GPa]	Flexural Strength [MPa]	Flexural Modulus [GPa]	Ref.
Jute/SR56 + SD	21.5	47	0.9	92	1.9	[110]
Jute/SR56 + SZ	29.0	73	8.0	115	7.2	[114]

## Data Availability

No new data were created or analyzed in this study. Data sharing is not applicable to this article.

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
