# Peer review of "Use of Bio-Epoxies and Their Effect on the Performance of Polymer Composites: A Critical Review"

_polymers, 2023, doi:10.3390/polym15244733_

Round 1

Reviewer 1 Report

Comments and Suggestions for Authors

The review report is attached

Comments on the Quality of English Language

In general, the article is easy to follow. However, some minor English grammar errors need to be revised. 

Author Response

Revision of manuscript Polymers-2761172

Camerino, December 4, 2023

Dear Reviewer 1,

We would like to thank you for your positive comments and for the suggestions.
In the following, please find the answers to your questions arised from the peer review are reported in blue.
In particular, all the changes are highlighted in yellow n the pdf version of the manuscript.

1. Please clarify the chemical building blocks mentioned in this review. (For example, Tannic acid under the category of tannoids.) Further clarification on the chemical constituents derived from lignin or food waste sources and their impact on the properties of bio-epoxy resins could enhance the article's depth and utility.
We apologize for this lack of information, since it was partially included in the first version (please, see Table 4). However, the chemical building blocks mentioned have now been added where missing,
and clarified in a better way where already present (please, see lines n. 196-199; 223-226; 463-465).
We hope that the new version of the manuscript will be aligned with the request. In particular, the chemical constituents extracted from lignin or food waste can lead to different impacts on mechanical and thermal properties, depending on the type of recycled material itself (as can be seen in Tables reported in the manuscript). In terms of sustainability, it is clear that a waste/recycled
material is preferable to a virgin synthesized one. Hence, since the raw materials are typically unknown, further research is needed in this area to better understand how the properties of bio-epoxy resins are affected by chemical constituents.

2. Please provide specific case studies or practical applications where bio-epoxy resins have been successfully utilized.
We have added Subsection 4.1 inside the “Proposed applications” Section, which describes the case studies of only bio-epoxy resins. In this way, the applications are split between the use of neat bio-epoxies and their presence in composite materials, according to the same philosophy of the entire manuscript (i.e., the two main bio-epoxies blocks of resins and composites).
3. Please include a section that discusses potential future directions for research, particularly regarding the optimization of bio-based content without compromising mechanical and thermal properties.
We have added the new “Outlook and challenges” Section before the Conclusions, which clarifies the requested aspects. Since the use of natural fibers in composite is increasing over the years, the study and the consequent use of bio-epoxy resins as matrix in composites are justified to achieve a
sustainable development. As mentioned above, the importance of extracting compounds from waste material is noted to everyone, instead of cultivating new ones. But, since their nature is usually unknown, an over-cured process was typically carried out to ensure the best properties. In particular,
the curing process reveals a major role in the final performances, along with the importance of monitoring the overall energetic impact of the process itself. So, the odds that are due to a non-virgin or contaminated material can significantly contribute to a decrease of the final properties. Hence, an optimization of the curing process – translated in the choice of appropriate parameters (such as time and temperature of exposure) – is preferable and needs to be investigated in the future.
4. Please revise minor English grammar errors in this article.
We thank the reviewer for the suggestion to improve the use of English language in our paper. The authors have gone through the text in order to correct the grammar errors.

Reviewer 2 Report

Comments and Suggestions for Authors

This review concern two aspects that the use of bio-epoxy resins and the performance of the composites made from bio-epoxy resins. Both of them are interesting topics for the scientific communities. There are some suggestions for the improvement of this reveiw paper.
Suggestions are as follows:
1.There are many reviews in the field of bio-epoxy resins, and some of them should be listed in reference to deliver more information to readers. e.g. reviews pubilshes in Chemical review, Progress in polymer science.

2. As for any kinds of epoxy resins, the composition should be epoxy resins and hardners at least, and fillers or reinforcements or not. Therefore,as for sustainability concerns, a question comes that those bio-epoxies resins cured by biobased harders or fossil based hardners. It is suggested that in any case the hardner should be pionted out they are biobased hardner or fossil based hardner.

3.For authors framework of this paper, it is clear but it better to define the scope of composites in the introduction part. For there are no information for nanocomposites in this review.

4. It is better to mentioned some progress in bio-based epoxy vitrimers, which is good approach for the sustainability composites. Therefore, it is could be an intereting direction for future for resins or composites.

Author Response

Revision of manuscript Polymers-2761172
Camerino, December 4, 2023

Dear Reviewer 2,

We would like to thank you for your positive comments and for the suggestions.
In the following, the answers to your questions arisen from the peer review are reported.
In particular, all the changes are yellow highlighted in the pdf version of the manuscript.
1. There are many reviews in the field of bio-epoxy resins, and some of them should be listed in reference to deliver more information to readers. e.g., reviews published in Chemical review, Progress in polymer science.
We thank the reviewer for the suggestion. In this revised version of the manuscript, we have included the references to some existing reviews on bio-epoxy resins.
2. As for any kinds of epoxy resins, the composition should be epoxy resins and hardeners at least, and fillers or reinforcements or not. Therefore, as for sustainability concerns, a question comes that those bio-epoxies resins cured by biobased hardeners or fossil-based hardeners. It is suggested that in any case the hardener should be pointed out they are biobased hardener or fossil-based hardener.
We regret that the bio-based or fossil-based nature of the hardeners is not clearly explained in the manuscript. In the first version, it was already included when bio-raw materials were into the epoxy system as hardeners, so the bio-nature of the hardener itself was implied (please, see lines n. 451, 503, 507, 624, 739, 756). If the epoxy resin provided other bio-hardeners, it was explicitly declared (please, see lines n. 207, 229, 403, 407, 479, 576); otherwise, we referred to a non-bio hardener, where its nature was omitted. However, the new updated version better clarifies this aspect (please, see lines 179- 181) and we hope that it will be aligned with the request.
3. For authors framework of this paper, it is clear but it is better to define the scope of composites in the introduction part. For there are no information for nanocomposites in this review.

We thank the reviewer for the comments. Since the use of natural fibers in composite materials is increasing over the years, the study and the consequent use of bio-epoxy resins as matrix in composites are giustified to achieve a sustainable development. Some information to better define the scope of composites are included in the introduction part (please, see the highlighted text from line 64). We agree with the reviewer that the nanocomposites are not cited in the work. Our research would like to point out the potentials of bio-based materials in applications where traditional composites (i.e., reinforced with synthetic fibers) are adopted. In this context, nanocomposites are out of the focus. However, we have added a reference to the use of bio-epoxy nanocomposites (Khaljiri, H.E.; Ghadi, A. Recent advancement in synthesizing bio-epoxy nanocomposites using lignin, plant oils, saccharides, polyphenols, and natural rubbers: A review. International Journal of Biological Macromolecules).
4. It is better to mention some progress in bio-based epoxy vitrimers, which is good approach for the sustainability composites. Therefore, it is could be an interesting direction for future for resins or composites.
We thank the reviewer for the suggestion to include some progress in bio-based epoxy vitrimers. As mentioned above for what concerns nanocomposites, also vitrimers are a bit out of the main purpose of the research, i.e. the state of the art in potential substitution of synthetic matrix or fiber in composites with natural fiber ones. Since it is already a long manuscript, the authors prefer to not go into detail about this aspect, except for a short mention of the state of art (please, see lines n. 337-347).

Reviewer 3 Report

Comments and Suggestions for Authors

1. Figure 1,3,6,8, are blurred.

2. line 540-541, "The use of MNA as bio-based curing agent, derived from furan and furfural, lead to a 540 completely bio-based epoxy resin [43]."  However, I checked the reference, the curing agent is MA. 

3. line 749 , there is a "?", I belive this is display error.

4. Table 3, EKL or DKL?

Author Response

Revision of manuscript Polymers-2761172

Camerino, December 4, 2023

Dear Reviewer 3,

We would like to thank you for your positive comments and for the suggestions.
In the following, the answers to your questions arisen from the peer review are reported.
In particular, all the changes are yellow highlighted in the pdf version of the manuscript.
1. Figure 1,3,6,8, are blurred.
We regret the quality of these images and hope that in the new version, the new figures are no longer blurred.
2. Line 540-541, "The use of MNA as bio-based curing agent, derived from furan and furfural, lead to a 540 completely bio-based epoxy resin [43]." However, I checked the reference, the curing agent is MA.
We thank the reviewer for the suggestion. In this revised version of the manuscript, we have corrected the typo on the abbreviation of curing agent.
3. Line 749 , there is a "?", I believe this is display error.
Perhaps this display error has occurred during the PDF generation of the .tex file. We hope that this issue has been resolved in the new version.
4. Table 3, EKL or DKL?
We appreciate the reviewer for pointing out this difference. The fourth row of Table 3 has been corrected in the new version.

Round 2

Reviewer 1 Report

Comments and Suggestions for Authors

I am happy with the revision.

Author Response

Thank you for your review and appreciation! 

Reviewer 2 Report

Comments and Suggestions for Authors

The authors made siganficant improvement in this revision, while some suggestions are listed as follows.

Suggestions for the paper.
1、In line 21, "These polymers consist of two components: base and thermoset which exothermically react. " in which "base and thermoset" are suggested to change into "hardner and resins".

2、For all the chemical structures, eg in Figure 1,3, 6,8, the bond length and size of benzene or other bond length are better in the same scale.

Author Response

We would like to thank you for your positive comments and for the suggestions.
In the following, the answers to your questions arised from the peer review are reported in blue.
In particular, the change within the text is yellow highlighted in the pdf version of the manuscript.

1. In line 21, "These polymers consist of two components: base and thermoset which exothermically react. " in which "base and thermoset" are suggested to change into "hardner and resins".
We thank the reviewer for the suggestion. In this revised manuscript, we have included the proposed change.

2. For all the chemical structures, eg in Figure 1,3, 6,8, the bond length and size of benzene or other bond length are better in the same scale.
We appreciate the reviewer for pointing out differences in scale. In the new version of the manuscript, the mentioned images now present the same scale for benzene and bond length.

Reviewer 3 Report

Comments and Suggestions for Authors

All question have been clearly answered. 

Author Response

(The authors gave the same response as above.)
